# Snowfall decrease in recent years undermines glacier health and meltwater resources in the Northwestern Pamirs
Achille Jouberton [1,2,3] ✉, Thomas E. Shaw [1], Evan Miles [3,4,5], Marin Kneib [6,7], Stefan Fugger [1,2,3], Pascal Buri [1,3,8], Michael McCarthy [1,3], Abdulhamid Kayumov [9], Hofiz Navruzshoev [9,10], Ardamehr Halimov [9,11], Khusrav Kabutov [9], Farrukh Homidov [9] & Francesca Pellicciotti [1]

Central Asia hosts some of the world's last relatively healthy mountain glaciers and is heavily dependent on snow and ice melt for downstream water supply, though the causes of this stable glacier state are not known. We combine recent in-situ observations, climate reanalysis and remote sensing data to force a land-surface model to reconstruct glacier changes over the last two decades (1999–2023) and disentangle their causes over a benchmark glacierized catchment in Tajikistan. We show that snowfall and snow depth have been substantially lower since 2018, leading to a decline in glacier health and reduced runoff generation. Remote-sensing observations confirm wider snow depletion across the Northwestern Pamirs, suggesting that a lack of snowfall might be a cause of mass losses regionally. Our results provide an explanation for the recent decline in glacier health in the region, and reinforce the need to better understand the variability of precipitation.

In High Mountain Asia (HMA), declines in water stored in glaciers and seasonal snowpacks have been widespread in recent decades[1,2]. Changes are however highly heterogeneous and have been attributed to differences in accumulation regimes[3,4], sensitivity to temperature increases[5,6] and regional differences in decadal precipitation changes[7]. In the semi-arid basins of the Pamirs, snow- and glacier melt sustain most of the annual streamflow[2,8,9], with glacier melt being especially important towards the end of the dry summers[10,11].

The Pamir Mountains host some of the only glaciers still featuring near-neutral mass balances[12,13]. The causes for this distinctive behavior are not fully understood, but increased snowfall[14,15], summer cooling, and reduced net energy have been suggested as potential explanations[16]. Recent evidence from remote-sensing observations, however, suggests that the period of relative stability may have transitioned since 2015[1,17], though this assessment is complicated by large spatial variability in glacier mass balance[18], the difficulty of identifying its dominant drivers[19] and the large uncertainties in the remote sensing estimates[1,12]. To investigate this possible transition, we combine hydrometeorological in-situ observations collected since 2021 at a glacierized

catchment in Tajikistan (Kyzylsu catchment), with downscaled ERA5-Land reanalysis and remote sensing observations to constrain a land-surface model applied at 100 m resolution from 1999 to 2023. Modeling can provide an understanding of mass balance seasonality and high-elevation catchment hydrology at a spatial and temporal resolution not afforded by observations, especially during the last two decades, for which little ground observations exist between the collapse of the Soviet Union and the re-establishment of monitoring networks[20–23]. The application of land-surface models has provided major advances in the recent understanding of energy and water fluxes of high elevation catchments in HMA, revealing for example the importance of evaporative fluxes and sublimation in these environments[11,24]. Accurate simulations, however, require robust estimates of snowfall amounts[25], which are challenging due to significant uncertainties in precipitation within reanalysis products[8,26–28], precipitation phase schemes[29], and in wind-driven undercatch on direct measurements[30].

We constrain snowfall estimates by leveraging simultaneous observations of near-surface meteorology, snow depth and snow water equivalent collected at a monitoring network established in 2021 at the Kyzylsu

---

[1]Institute of Science and Technology Austria, ISTA, Klosterneuburg, Austria. [2]Institute of Environmental Engineering, ETH, Zürich, Switzerland. [3]Swiss Federal Research Institute WSL, Birmensdorf, Switzerland. [4]Glaciology and Geomorphodynamics Group, Department of Geography, University of Zürich, Zürich, Switzerland. [5]Department of Geosciences, University of Fribourg, Fribourg, Switzerland. [6]Institut des Géosciences de l'Environnement, Université Grenoble-Alpes, CNRS, IRD, Grenoble, France. [7]Department of Atmospheric and Cryospheric Sciences, University of Innsbruck, Innsbruck, Austria. [8]Geophysical Institute, University of Alaska Fairbanks, Fairbanks, AK, USA. [9]Center for the Research of Glaciers, Tajik National Academy of Sciences, Dushanbe, Tajikistan. [10]Mountain Societies Research Institute, University of Central Asia, Dushanbe, Tajikistan. [11]State Key Laboratory of Ecological Safety and Sustainable Development in Arid Lands, Xinjiang Institute of Ecology and Geography, Chinese Academy of Sciences, Urumqi, China. ✉e-mail: achille.jouberton@gmail.com

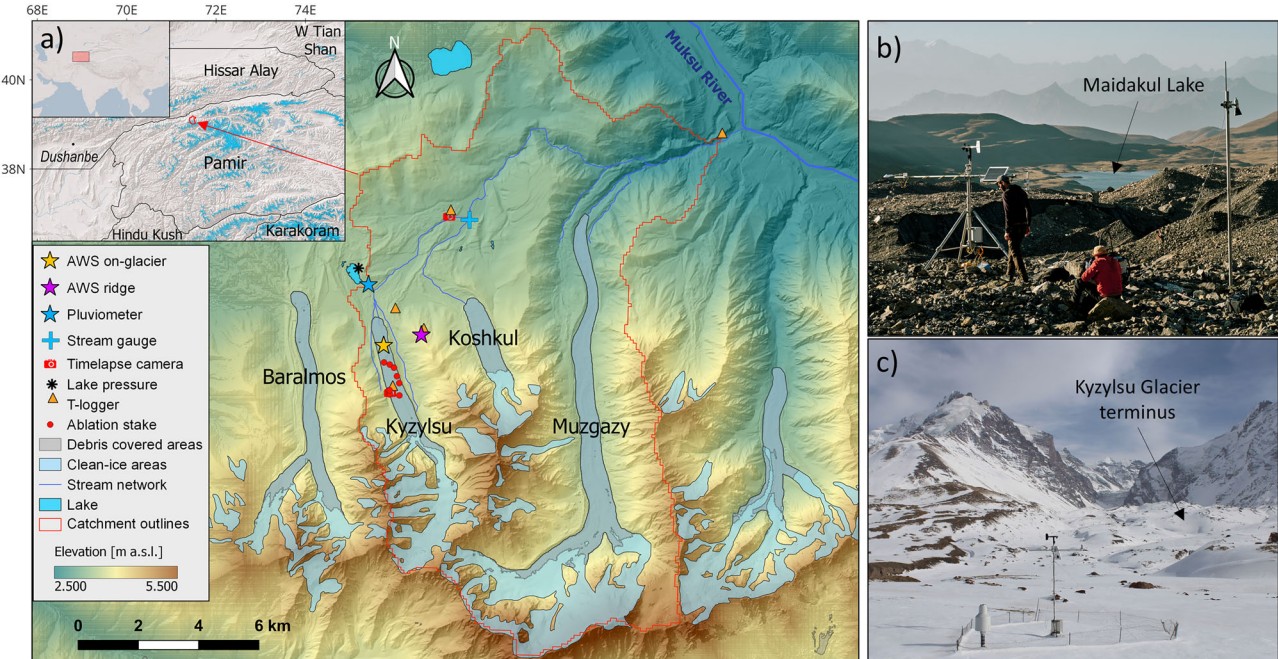

**Fig. 1 | Study site and its monitoring network. a** Map of the Kyzylsu catchment. The names of the main glaciers are indicated in black. The elevation information is taken from the AW3D Digital Elevation Model (DEM), while the hillshade was derived from high-resolution Pleiades DEMs acquired in 2022 and 2023. Glacier outlines and debris extents are from the RGI 6.0 inventory. Lakes were manually delineated from a Pléiades 2022 ortho-image. The inset maps show the location of the study site in Central Asia with a base map from Esri, along with glaciers shown as blue areas and sub-regions outlines from the RGI 6.0 inventory. **b** Picture taken by Jason Klimatsas in September 2023 of the on-glacier automatic weather station, located on the debris-covered portion of Kyzylsu Glacier. Maidakul Lake can be seen in the background, as indicated by an arrow. **c** Pluviometer station photographed by a time-lapse camera in March 2022, with the snow-covered terminus of Kyzylsu Glacier visible in the background. Photos of all other hydrometeorological stations are shown in Supplementary Fig. S1.

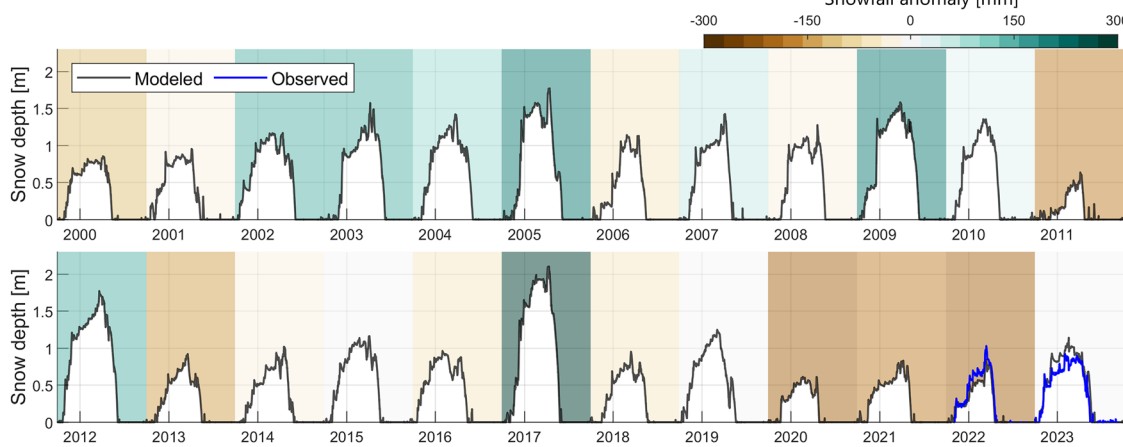

**Fig. 2 | Snow depth reconstruction since 1999.** Simulated daily snow depth at the pluviometer station (3369 m a.s.l.) since October 1999 (in black) and measured from 20/09/2021 to 30/09/2023 (when data are available, in blue). The colored shaded background indicates the annual snowfall anomaly of the corresponding hydrological year (from October 1st to September 30th).

catchment (Fig. 1), Tajikistan. We apply a detailed energy-balance land-surface model from 1999 to 2023, evaluated against in-situ and remotely sensed observations (Supplementary Table S2), to demonstrate that snowfall and snow cover have been substantially lower since 2018, leading to enhanced ice melt and affecting glacier mass balance and health. The snowfall decline was driven by a decrease in total precipitation, while its partition (snow versus liquid amounts) remained largely unchanged. The unbalanced ice melt resulting from a lack of snow compensated for a third of the precipitation-driven catchment runoff deficit, but accelerated glacier demise. We investigate the validity of our results beyond the study catchment and find that remote-sensing observations and climate reanalysis

confirm the decline in snow cover and precipitation also for the wider Northwestern Pamirs region.

## Results

### Reduced snowfall, snow depth and snow cover since 2018

At the pluviometer station (Fig. 1), we reconstruct a snow depth time series since October 1999 (Fig. 2), revealing that peak snow height has been substantially lower since 2018 ($0.92 \pm 0.25$ m) relative to 1999–2018 ($1.30 \pm 0.37$ m, Fig. 2), associated with an earlier snowpack melt-out in spring (by around two weeks, Supplementary Fig. S33). Unless stated otherwise, the range of values given along with numerical results indicates variability (1

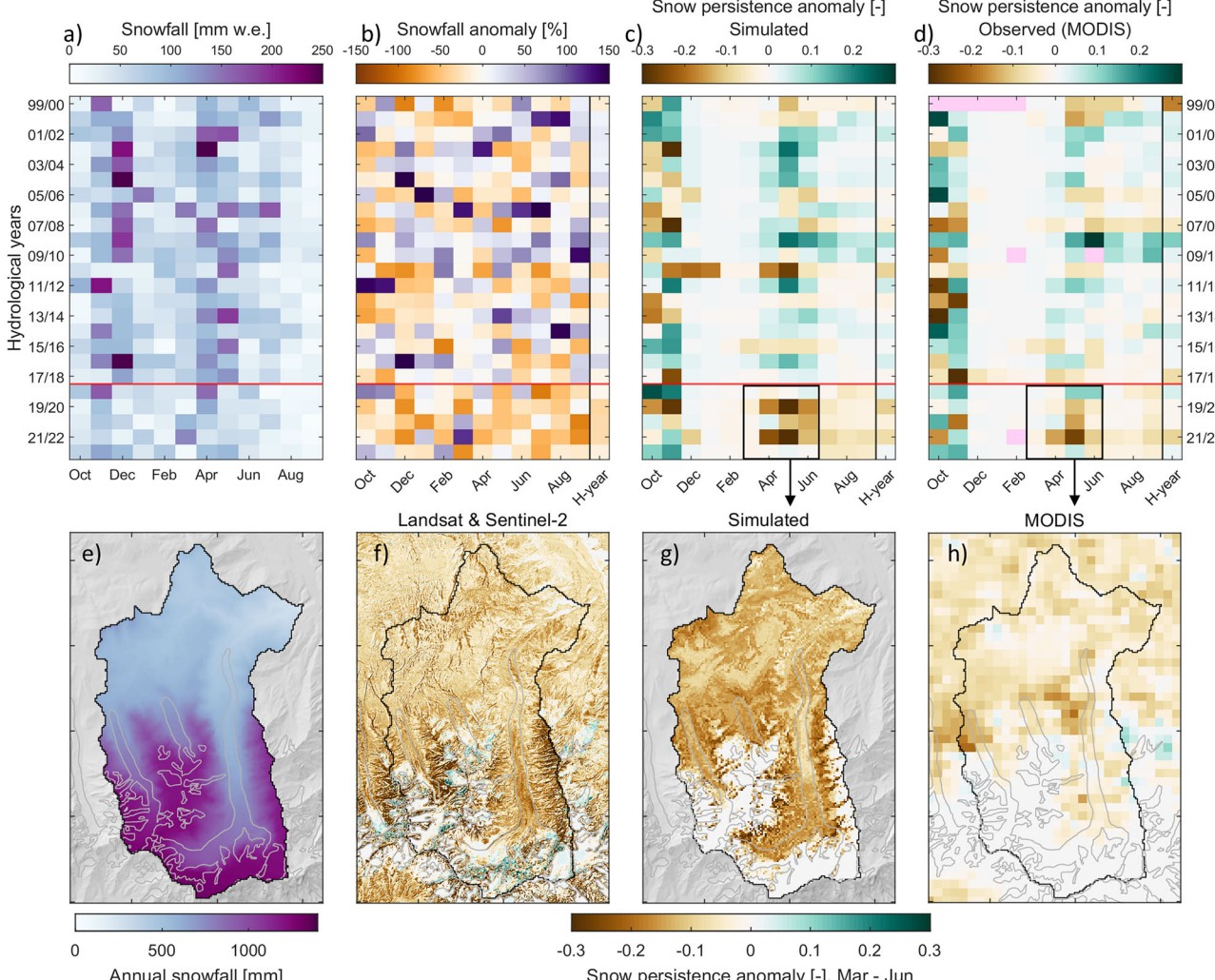

**Fig. 3 | Decline in snowfall, snow depth and snow cover at Kyzylsu catchment.**
**a** Mean monthly snowfall amounts averaged over the catchment area. **b** Mean monthly snowfall anomaly relative to the mean of all years in a given month. **c** Simulated snow persistence anomaly and **d** snow persistence anomaly derived from the MODIS daily snow cover product, where pink-colored pixels indicate an absence of data for the corresponding month. H-year in panels (**b–d**) refers to the hydrological year (October 1st to September 30th). The horizontal red line shown in

(**a–d**) separates the first (1999–2018) and the second, drier sub-period (2018–2023). **e** Simulated annual snowfall average over the whole period (1999–2023). **f–h** Changes in mean snow persistence between 2018–2023 and 1999–2018 for the months of March to June observed by Landsat 5/7/8/9 and Sentinel-2 imagery (**f**), simulated by the land-surface model (**g**) and observed by MODIS (**h**), using only dates for which Landsat 5/7/8/9 or Sentinel-2 scenes are available and cloud-free. Glacier outlines shown as gray lines in (**e–h**) are from the RGI 6.0 inventory.

standard deviation). Annual snowfall amounts have also been mostly below average since 2018 (Fig. 2): mean annual snowfall (total precipitation) at the station decreased from 617 (1176) mm yr$^{-1}$ in 1999–2018 to 470 (811) mm yr$^{-1}$ in 2018–2023. Precipitation falls all year round (Fig. 3a), with higher amounts in spring (April to June) and early summer (July) when convective precipitation events are infrequent but can be intense (e.g. 31 mm h$^{-1}$ measured on July 16th 2023), while the driest conditions are occurring in August and September (together < 10% of the annual total). Monthly anomalies in catchment-wide snowfall between 1999–2018 and 2018–2023 suggest that the snowfall decrease has happened throughout the year (Fig. 3b). Snow cover has been less persistent across the catchment in the recent period from 2018 onwards, especially during spring (Fig. 3c). This is consistent with the results from cloud-free scenes of the MODIS (Moderate Resolution Imaging Spectroradiometer) daily snow cover product (Fig. 3d, h) and higher resolution optical imagery (Landsat 5/7/8/9 and Sentinel-2) which can better map high elevation regions of ice and snow (Fig. 3f). Correlations between the simulated snowpack melt-out date and the reconstructed climatic forcing (Supplementary Fig. S41) show that the earlier seasonal snowpack melt-out from 2018 onwards was caused

primarily by lower winter snowfall, leading to thinner winter snowpacks, and secondarily by warmer spring air temperatures (+0.19 °C, Supplementary Fig. S36).

## High-elevation snowfall decline has driven recent glacier mass loss

The decrease in snowfall between 1999–2018 and 2018–2023 was mainly caused by a 328 mm decrease (−28%) in total precipitation, and was more pronounced above 4000 m above sea level (a.s.l., Figs. 3e and 4a), where the solid precipitation ratio (hereafter snowfall fraction) is higher (Fig. 4b). Snowfall changes cannot be attributed to a change in the annual snowfall fraction, which remained stable between the two periods (Fig. 4b), but rather to precipitation changes (Supplementary Fig. S43). This stable snowfall fraction is found despite a slight decline in the summer snowfall fraction (Supplementary Fig. S37), as the decrease in summer precipitation is larger than for the other seasons (Supplementary Fig. S38).

The reconstructed annual glacier mass balance of Kyzylsu Glacier since 1999 (Fig. 5) shows that snowfall contributes to mass inputs in all seasons, with greater amounts in April to June (379 mm/37% of snowfall total),

**Fig. 4 | Simulated annual snowfall, snowfall fraction and area fraction per elevation. a** Snowfall per elevation for each hydrological year (thin lines) and averaged over the 1999–2018 and 2018–2023 periods (thick lines). **b** Same as (**a**) but for the snowfall fraction, i.e. the fraction of total precipitation that falls in the solid form. The hypsometry of the catchment and its glaciers are shown as gray and blue shaded areas, respectively.

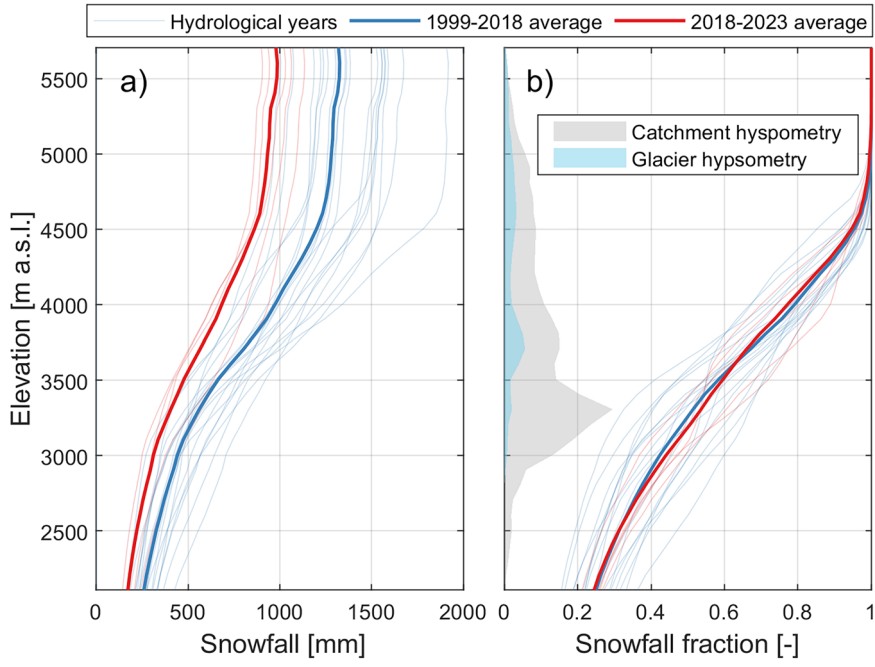

followed by October to December (276 mm/26%). The decrease in on-glacier snowfall since 2018 was most pronounced in July to September (−120 mm/−50% relative to the mean of 1999–2018) followed by April to June (−100 mm/−25%), contributing to a strong decline in glacier health, with the mean glacier mass balance decreasing from −0.11 ± 0.40 m w.e. yr$^{-1}$ in 1999–2018 to −0.82 ± 0.26 m w.e. yr$^{-1}$ in 2018–2023. The snowfall reduction contributes to a substantial part (−0.31 m w.e. yr$^{-1}$) of the additional glacier mass loss (−0.71 m w.e. yr$^{-1}$), which was also due to enhanced ice melt (+0.32 m w.e. yr$^{-1}$). However, the snowfall decline contribution to this additional mass loss could be higher due to its role in the decrease of snow cover and surface albedo (−0.023, Supplementary Fig. S45), which in turns contributes to higher ice melt, and which is difficult to quantify due to the concurrent effect of warmer summer temperatures (+1.2 °C, Supplementary Fig. S36) on snow cover duration, albedo and ice melt rates. We find a strong correlation between modeled annual snowfall and glacier mass balance (R$^2$ = 0.92, Supplementary Fig. S42). July 2022 was the warmest month in the observed period at the on-glacier weather station (3579 m a.s.l.), with a mean (maximum) air temperature of 10.4 (21.8) °C, during which time record-breaking temperatures, above 43 °C, were recorded in the plains of Tajikistan (Supplementary Fig. S44). Over that month, the simulated mass balance of Kyzylsu Glacier was −0.66 m w.e., far more negative than the 1999–2023 July average (−0.29 ± 0.18 m w.e.), highlighting the sensitivity of these glaciers to summer temperatures (R$^2$ = 0.51, Supplementary Fig. S42). Vapor fluxes (evaporation and sublimation) contributed to the glacier mass loss by 0.16 m w.e. yr$^{-1}$ (representing about 16% of the annual snowfall input) and barely changed (+3%) between the two sub-periods. Avalanches, however, supplied much less snow: 0.08 ± 0.05 m w.e. yr$^{-1}$ of mass (glacier-wide rate) from the surrounding headwalls in 2018–2023, compared to 0.21 ± 0.10 m w.e. yr$^{-1}$ in 1999–2018, slightly enhancing the additional mass losses caused by the reduction of direct snowfall on the glaciers (−0.31 m w.e. yr$^{-1}$). Our simulations suggest that little net mass accumulation occurred during 2018–2023 (Fig. 6), which is consistent with the Pleiades satellite image of 24 September 2022 where snow on the glaciers is only visible at the highest elevations (Supplementary Fig. S13).

**Can ice melt compensate for precipitation-driven runoff deficits?**
Analysis of the catchment water fluxes shows that the decline in snowfall and total precipitation is associated with an overall decrease in annual runoff at the catchment scale (−189 mm w.e. between 1999–2018 and 2018–2023, Fig. 7).

The runoff deficit is however smaller than the precipitation deficit (−363 mm) because 31% of it was offset by glacier meltwater. The glacier melt contribution to runoff increased from 19% to 31% between the two sub-periods, while the relative importance of snowmelt did not change as much (from 50 to 46%). In absolute terms, the decrease in annual snowmelt (−132 mm) is larger than the increase in ice melt (+86 mm), leading to an overall decrease in meltwater (−46 mm). These changes in runoff contribution had consequences on the seasonality of net runoff (Fig. 7e, f). The decline in spring snowmelt and rainfall on the wide grassland plateau (3000–3500 m a.s.l.) led to less net runoff during the month of May, which was partially compensated by an increase in ice melt from July to September (at 3600–4000 m a.s.l., Fig. 7a–c). The reduction in precipitation in the last five years has therefore amplified the role of frozen water storages located above 3600 m a.s.l.. Despite lower amounts of precipitation, evapotranspiration slightly increased (+9 mm/+4%, Fig. 7c) resulting in a decrease in the runoff ratio (from 0.75 to 0.70) and a higher catchment runoff deficit. Warmer summer conditions have enabled more water to evaporate over the plateau, while a decrease in ET is simulated near the catchment outlet (Supplementary Fig. S40) due to reduced water availability.

## Discussion

The unique observations at the Kyzylsu glacierized catchment allowed us to constrain the meteorology at this site and run a land-surface model at high spatial and temporal resolution. We evaluated the model using both ground and remotely sensed observations, and use it to explain the recent decline in glacier mass balance suggested by remote sensing studies. We report a decline in snow depth, snow cover, glacier mass balance, and total runoff over the most recent decade, which is especially strong in 2018–2023 relative to 1999–2018. This is, to the best of our knowledge, the first report of declining snow cover and snowfall in recent years in the region, as most recent and large-scale studies do not have yet a period of analysis extending beyond 2020[31]. The application of our modeling framework at the regional scale is challenging due to the strong spatial variability of meteorological conditions within this mountainous region[27] and the general lack of ground observations to bias-correct reanalysis products and evaluate model performance, notwithstanding the computational costs involved with regional-scale simulations containing this level of process representation[11,32]. Regional anomalies in snow cover derived from MODIS observations and total precipitation from ERA5-Land reveal, however, that this decline in snow cover and precipitation since 2018 has affected a majority of the Northwestern Pamir region (Fig. 8).

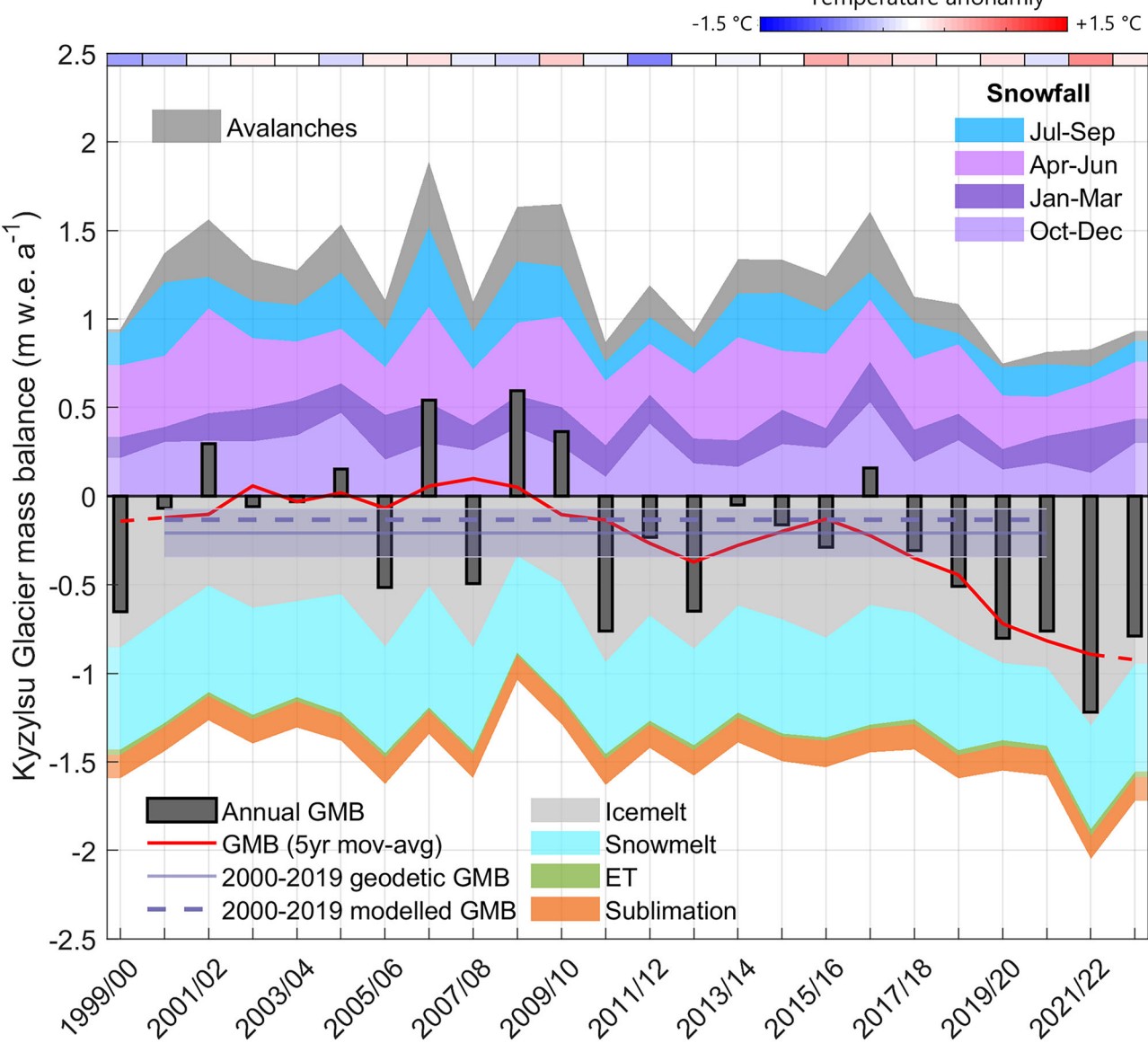

**Fig. 5 | Mass balance changes at Kyzylsu Glacier simulated since 1999.** Shaded areas indicate annual time series of mass inputs (seasonal snowfall, avalanches) and mass losses (icemelt, snowmelt, evapotranspiration, sublimation). These mass balances components add up to the net glacier mass balances (GMB), shown as vertical bars for each hydrological year and as a red line for a 5-year moving average. The 2000–2019 geodetic mass balance shown as a solid purple line and its uncertainty shown as a shaded area are from Hugonnet et al.[1]. Mean annual air temperature deviations from the period average are shown on top of the figure as colored rectangles.

The actual beginning of the decline in snowfall has some ambiguity. We selected the end of the hydrological year 2018 (September 30th) as a middle point for our two sub-periods of analysis based on the largest changes in mean annual precipitation and glacier mass balance (Supplementary Fig. S34), yet snowfall amounts have generally been below average since 2012 (Fig. 2). We find the same decline in all variables described above using 2012 as a middle point, yet with a less marked anomaly in snow cover and precipitation (Supplementary Fig. S35), indicating that both climatic and hydrological conditions have changed most strongly in very recent years. The drivers behind the drier conditions detected in 2018–2023 remain unknown, and could be linked to large-scale forcing and teleconnections such as the Pacific Decadal Oscillation[33], El Niño-Southern Oscillation and North Atlantic sea surface temperature anomalies[34]. The northwestern Pamirs generally receive moisture from the mid-latitude western disturbances[35,36], but changes in their frequency and intensity are not fully understood despite recent advances in analysis techniques such as tracking algorithms and the increasing availability of high-resolution weather and climate models[37]. The dry conditions observed in recent years might not persist in the future, as projections from

the Coupled Model Inter-comparison Project Phase 6 (CMIP6) generally agree on an increase in precipitation in Central Asia by the end of the 21st century[38,39], despite large differences among climate models[7] which call for further efforts to understand the drivers and mechanisms of precipitation variability in the region.

A major source of uncertainty in land-surface and glacio-hydrological models at high elevations relates to the meteorological forcing and its spatial variability that is seldom captured by in-situ station networks or by often too coarse atmospheric model outputs[28,40,41]. Our model was able to reproduce observed snow-line elevation dynamics (Supplementary Figs. S29–32) and geodetic mass balances (Fig. 5) since the year 1999 without calibrating a precipitation vertical gradient or correction factor unlike what is usually necessary[24,42–44]. This also provides an indirect validation of our reconstructed precipitation, which cannot be directly evaluated outside of the observation period used for bias correction (i.e. before 2021). The correction of precipitation measurements for wind-driven undercatch using snow water equivalent derived from lake pressure (cf. Methods) was key to obtaining reasonable snow depth simulations (Supplementary Fig. S3), and

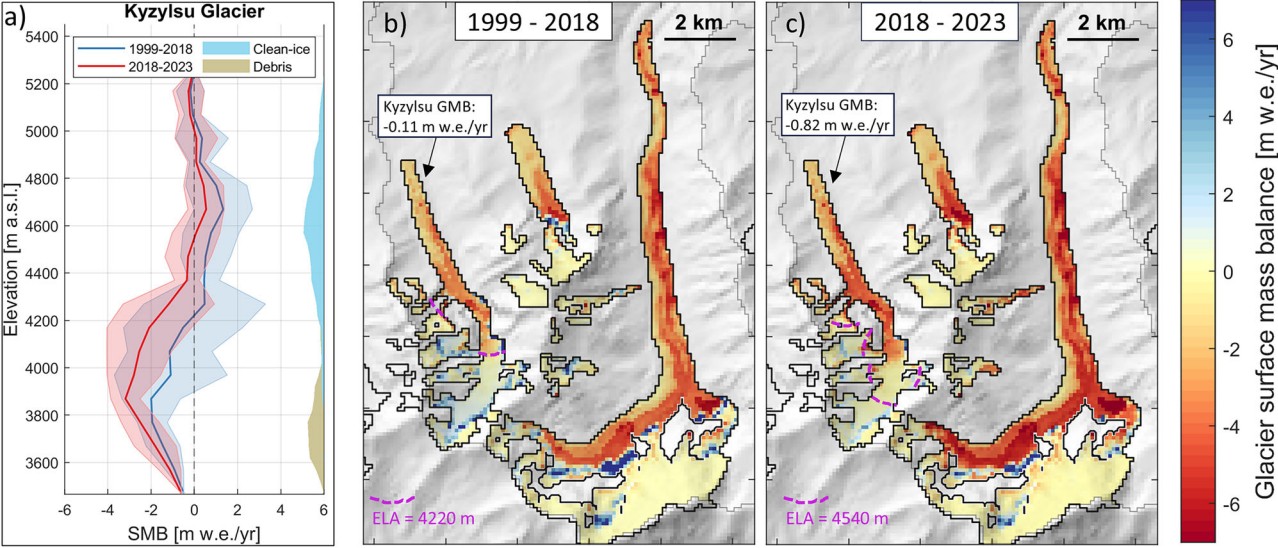

**Fig. 6 | Surface mass balance of glaciers simulated at Kyzylsu catchment. a** Surface mass balance per 100-m elevation band for Kyzylsu Glacier, with shaded areas indicating one standard deviation. The glacier hypsometry (in km²) is shown on the right of the panel. Spatially distributed surface mass balance for the 1999–2018 (**b**) and 2018–2023 (**c**) periods. The mean glacier-wide surface mass balance (SMB) of Kyzylsu Glacier is indicated in a textbox, while the equilibrium line altitude (ELA) is drawn as a dashed purple line. The hillshade information shown in the background of (**b**, **c**) is taken from the AW3D DEM resampled to the spatial resolution of the model simulations (100 m).

we find that the use of uncorrected precipitation measurements can lead to worse model performance than when using the raw ERA5-Land precipitation product (Supplementary Fig. S3). Crucially, the bias correction of ERA5-Land precipitation led to a marked change in its seasonality (Supplementary Fig. S8) and therefore in its interannual variability; our precipitation measurements indicate relatively higher amounts in June and July compared to what is estimated by ERA5-Land. From repeated field visits in early summer as well as timelapse camera records, it is clear that local convective precipitation events occur frequently. While these summer precipitation events may be driven by the recycling of water from land surface evaporation[45], they are often not captured by reanalysis models due to their coarse resolution. Understanding the seasonality of precipitation is key to assessing the impact that air temperature increase has and will have on the future of the precipitation phase and related glacio-hydrological processes (e.g. surface albedo, ice snow and ice melt, runoff). High-quality and continuous in-situ observations of all-phase precipitation are therefore paramount to understanding and quantifying the precipitation regimes of such under-studied and largely ungauged mountain catchments. Our pluviometer station has operated at 3369 m a.s.l. in Tajikistan since 2021 and contributes to filling the regional monitoring gap in a country where more than half of the land is located above 3000 m elevation.

The future of the region's glaciers and of the water resources they sustain is the focus of an intense debate and considerable research[15,16,22]. Our simulations reveal that changes in snowfall were the largest in glacier accumulation areas due to a high solid precipitation ratio and, therefore, higher sensitivity to total precipitation changes. The sensitivity of mass balance to precipitation changes was enhanced by the supply of mass through the headwalls of Kyzylsu Glacier (Figs. 5 and 6), and we found that simulations that do not include gravitational redistribution would result in a lower interannual variability of mass balance (Supplementary Fig. S23). Projected long-term (2081–2100) changes in temperature suggest a warming of 3 °C under SSP2-4.5 and 6 °C under SSP5-8.5 relative to 1995–2014[39], which will affect the phase of precipitation at high elevations[46] and could prevent further periods of snowfall increase driven by increases in total precipitation. Our simulations show no change in snowfall fraction between 1999–2018 and 2018–2023, but substantial changes in snowfall fraction can be expected from March to October by the end of the 21st century, months for which the air temperature is not largely below 0 °C (Supplementary Figs. S36 and S39), and their consequences on glacier mass

balance and runoff seasonality should be investigated. In general, a substantial effort is still needed to understand the current state of the Pamir's glaciers and their controlling factors, and this study provides an initial step into understanding the recent decline in glacier health and its causes. Longer term reconstructions of snow and glacier mass balances are needed to further understand the long-term sensitivity of these glaciers to temperature and precipitation changes, and project their future with confidence.

## Methods
### Study site
The Kyzylsu catchment (168 km²) lies on the northwestern slopes of the Pamir mountain range in Tajikistan, with its outlet located at 2100 m a.s.l. It serves as a tributary to the Vakhsh River, downstream of which the Roghun Dam-one of the tallest in the world-is currently under construction, a project expected to substantially increase Tajikistan's hydropower energy production[47]. The climate is continental and semi-arid, influenced by mid-latitude westerlies and local moisture recycling[48]. Most of the precipitation falls in winter and spring[11,36], and the glaciers are considered as winter-accumulation type[4]. The catchment ranges in elevation from 2110 m a.s.l. to 5806 m a.s.l.. The predominant land covers comprise pastures (50%), extensively utilized for grazing by local livestock, glaciers (20%), several lakes, and rocky surfaces (29%). The glaciers are predominantly debris-covered, have experienced surges in the last 30 years, and have their surfaces level with their lateral moraines in response to the limited mass losses in recent decades. We established a monitoring network (Fig. 1) in June 2021 consisting of several automatic weather stations (AWSs), one of which includes an all-type precipitation weighing gauge (referred to as pluviometer station), time-lapse cameras, lake and stream level gauges, glacier ablation measurements, and air temperature loggers. Continuous snow depth time series were retrieved at four locations, either from daily camera photos or ultrasonic depth gauges. Supplementary Table S1 summarizes the instruments' locations, variables recorded, and recording periods. The data collected from June 2021 to September 2023 were used to downscale and bias-correct ERA5-Land reanalysis[49,50] and evaluate the model's performance.

### Near-surface meteorology
In-situ measurements of near-surface meteorology were conducted to better characterize the climate of this high-elevation site and produce the time series needed for the statistical downscaling and bias correction of the ERA5-Land

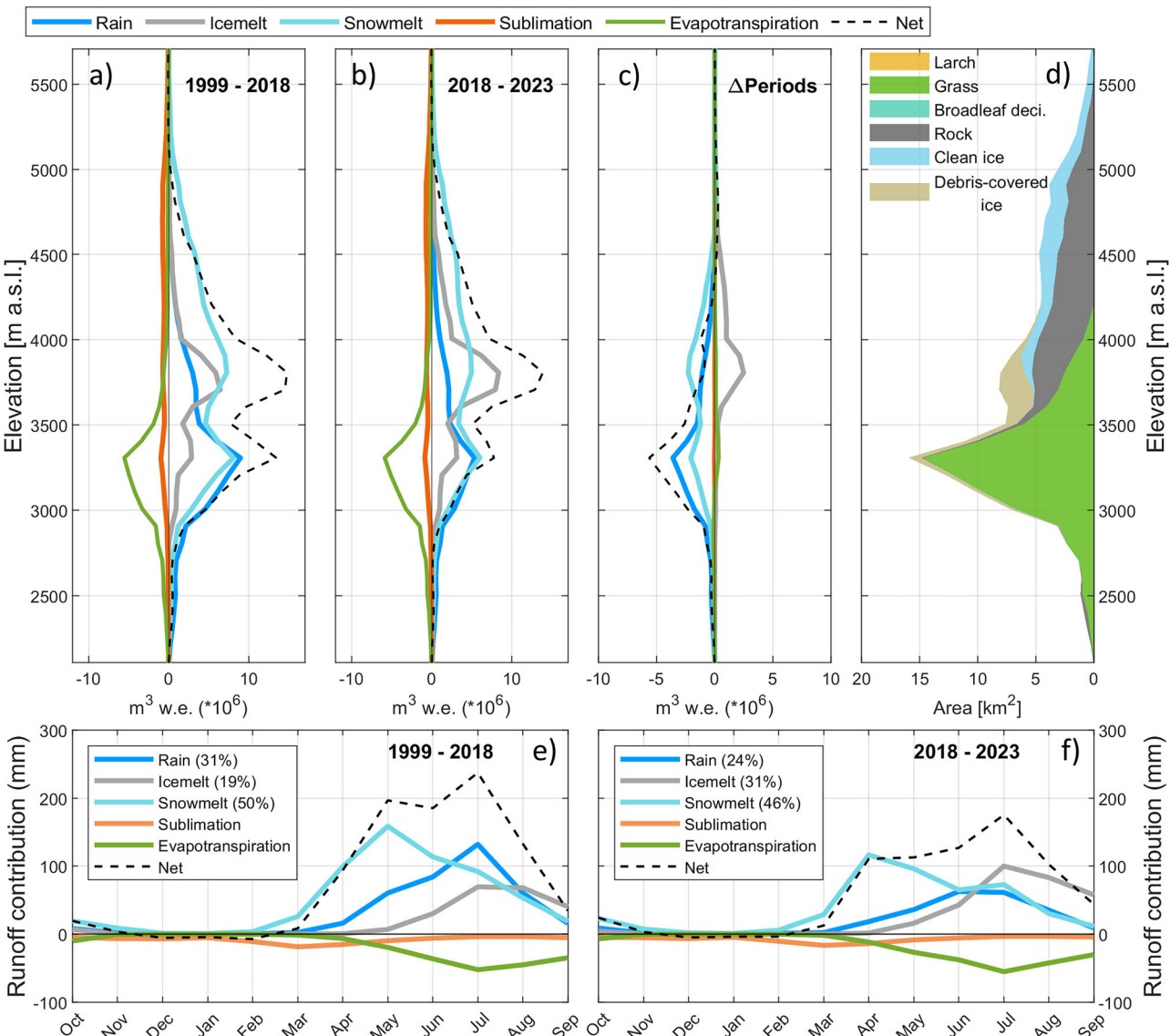

**Fig. 7 | Water fluxes and contributions to runoff generation at Kyzylsu catchment.** Water fluxes averaged per elevation band over the periods 1999–2018 (**a**) and 2018–2023 (**b**). **c** Changes in water fluxes per elevation band between the two sub-periods. **d** Catchment hypsometry used to convert the altitudinal fluxes into volumes in (**a**–**c**), with shaded areas corresponding to different land cover types. **e**, **f** same as (**a**–**c**) but averaged per calendar month over the catchment area. The net runoff shown as a black dotted line is the sum of all the other variables displayed. The numbers (in %) given in the legends of (**e**, **f**) correspond to the relative contribution to the total runoff generation.

reanalysis. The air temperature was measured at various locations in the catchment (Fig. 1a) from 2021 to 2023 at hourly intervals, which we used to derive hourly air temperature lapse rates, later used in the downscaling of ERA5-Land reanalysis. The derived lapse rates show clear seasonal and diurnal cycles, consistent from one year to the other (Supplementary Fig. S2). One of the AWSs (pluviometer station) is equipped with an OTT all-weather precipitation gauge and recorded precipitation continuously from September 2021 to September 2023. A windshield was only added to the pluviometer in September 2023, such that the precipitation data used in this study was recorded without a windshield. To correct for undercatch, we used the empirical relationship established by Masuda et al.[51], which is given below:

$$P_{corr} = P_{obs} \cdot (1 + mU) \tag{1}$$

Where $U$ is wind speed (m/s) at the height of the rain gauge and $m$ is an empirical correction coefficient which depends on the precipitation type and the type of rain gauge. The empirical coefficients of this

relationship were derived in Japan, with different climatic characteristics than our Central Asian study site, so we estimated the correction factor for solid precipitation using cumulative snow water equivalent derived from lake pressure measurements conducted from September 2021 to June 2022. Maidakul Lake is located 500 m away from the pluviometer station and freezes over during the winter. We followed the approach described in Pritchard et al.[52], to convert increases in water pressure into snow water equivalent of snowfall (in mm w.e.). We then calibrated the pluviometer snowfall correction coefficient ($m$) to match the total accumulated snowfall amount estimated between November 2021 and February 2022, a period during which the lake surface was frozen, and the air temperature was well below 0 ˚C (Supplementary Fig. S3). We obtained a correction coefficient for solid precipitation of 0.30, which is close to the empirical value given (0.34) in the original study[51]. We further validated the choice of the precipitation undercatch correction coefficient by conducting snow depth simulations using the different options of precipitation undercatch correction ($m = 0$, $m = 0.30$, $m = 0.34$) (Supplementary Fig. S3). Shortwave radiation (incoming and

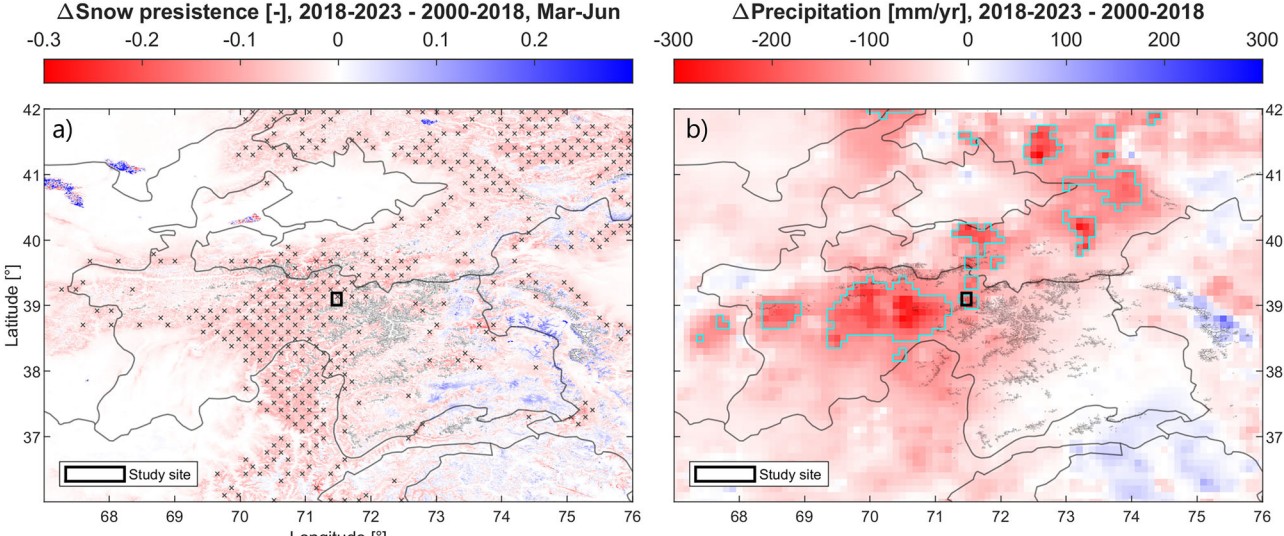

**Fig. 8 | Regional anomalies in precipitation and snow cover, comparing 2000–2018 and 2018–2023. a** Anomaly in mean snow cover persistence derived from the daily MODIS snow cover product (MOD10A1.061), focusing on the March to June period. Areas experiencing a snow cover anomaly similar to or larger than our study site are indicated by black crosses. **b** Regional anomaly in annual precipitation from ERA5-Land. Areas experiencing a precipitation anomaly similar to or larger than our study site are outlined in light blue. World administrative boundaries (countries and territories) are indicated as black solid lines, and glaciers from the RGI 6.0 inventory are shown as light gray solid lines. Our study site is located at the center of the map and is indicated by a black box.

outgoing), wind speed, air pressure, and relative humidity were recorded at the pluviometer station and at the on-glacier AWS, while the latter also recorded longwave radiation (incoming and outgoing).

### Snow depth observations
Snow depths were continuously monitored at the two AWSs using ultrasonic depth gauges and at two additional locations using time-lapse cameras with graduated vertical stakes set in their field of view. For the camera-derived snow depth, the snow height was manually read from daily pictures and an uncertainty of $+/-5$ cm was attributed to each snow depth value[53]. The measured snow depths were used to evaluate the model performance and to assess the effect of the precipitation undercatch correction (Supplementary Figs. S3, S27).

### High-resolution snow cover maps
We derived snow cover maps from MODIS, Landsat 5/7/8/9, and Sentinel-2 to investigate the spatiotemporal changes of the catchment snow cover and evaluate our model performance against observed fraction snow cover and snow line altitude. We used the MODIS daily product MOD10A1 version 6.1 and identified snow-covered areas using a Normalized Difference Snow Index (NDSI) threshold of 0.40, but also with thresholds of 0.25 and 0.45 to quantify the uncertainty[54]. We discarded all MODIS scenes having a cloud cover fraction greater than 0.10 to reduce the risks of cloud/snow misclassification[55]. While the temporal resolution of MODIS is very high (daily), its spatial resolution (500 m) is rather coarse relative to the scale of our study domain, especially in the upper areas of the domain which have high spatial variability of slope, aspect, and surface conditions. We therefore also used all available cloud-free Landsat-5/7/8/9 and Sentinel-2 scenes to derive snow cover maps of higher spatial resolution (30 m). Outside of clean-ice glacier areas, we converted multi-spectral surface reflectance into a binary snow presence indication. This method[56], based on NDSI and reflectance in the red band, does not perform well for discriminating bare ice from snow, such that over clean-ice areas we used surface albedo derived from the surface reflectance[57] to distinguish snow and bare ice[58]. The critical albedo threshold was determined for each scene using the Otsu algorithm, bounded by 0.25 and 0.55 and assuming a bi-modal distribution of albedo values on glacier areas[58]. Examples of the snow cover mapping are provided in the Supplementary Material (Supplementary Figs. S11, 12).

### Spatially resolved snow depth from very high-resolution stereo imagery
Digital Elevation Models (DEMs) derived from high-resolution (<5 m) optical stereo images (e.g. Pléiades) can be used to derive high-resolution snow depth maps over entire catchments[59]. We acquired two Pléiades stereo images over the Kyzylsu catchment and derived DEMs (2 m spatial resolution) using the Ames Stereo Pipeline[60]. The first scene was acquired on September 24th 2022 during snow-free conditions, and the second one was acquired on May 23rd 2023 when a large proportion of the catchment was snow-covered. We performed co-registration[61] and corrected for tilt and along-track undulations[62] (Supplementary Fig. S14). Co-registration and DEM differencing were performed using the open-source Python package xDEM[63]. We used the surface elevation change to identify the location of avalanche deposits and qualitatively evaluate the simulated gravitational redistribution of snow.

### Land-surface model
The Tethys-Chloris model[32,64] is an ecohydrological model simulating the interplay between energy, water, and vegetation dynamics on the land surface. It has recently been updated to include energy-balance calculations over snow, clean-ice, and debris-covered glaciers[65] and simulate the water balance of high-elevation catchments[11,24]. The model explicitly simulates lateral water transfer, surface runoff, and subsurface flow, capturing the complex dynamics of soil moisture and its interactions with vegetation. It uses a single prognostic surface temperature to calculate energy fluxes and determines aerodynamic resistances for turbulent heat and vapor fluxes using the Monin-Obukhov Similarity Theory. The partitioning of precipitation into rain, snow and sleet is based on the wet bulb temperature[66], the snow albedo dynamics are simulated using the parameterization of Ding et al.[66] and the model also includes schemes for snow aging, snow settling and surface sublimation[32]. It uses a 2-layer snowpack model, featuring a 6-mm thick surface skin layer that enables energy exchange with the atmosphere and heat transfer within the snowpack. While heat transfer is calculated across multiple layers, the model simplifies computations by using single prognostic variables for snow density and water content. Liquid water can infiltrate and be stored in the snowpack, within a volume given as a percentage of the total volume, and can refreeze depending on the latent heat flux. Snow is converted to ice at a constant rate once the snowpack reaches a threshold density. The model does not include a firn layer, but this

should not affect our main results (cf. Supplementary Section 4.5 in Supplementary Material). The model uses the SnowSlide parametrization[67] to route avalanches as a function of slope and maximum snow holding depth. Glacier flow was not accounted for in this study, as glacier areas and glacier surface elevation did not change substantially through that period (Supplementary Fig. S19). The surface elevation was therefore kept constant for the downscaling of the meteorological forcing, and the ice thickness, based on the ice thickness consensus estimate[68], was increased artificially to prevent the disappearance of the ice column in the ablation zone. We use the globally available AW3D Standard DEM[69] resampled to 100m as a reference surface elevation to downscale the meteorological forcing, calculate runoff routine, delineate the catchment outline, and avalanche flow paths. The model is run at an hourly time-step from 1 October 1999 to 30 September 2023. In this study, we define the hydrological year as starting on October 1st and ending on September 30th and use this as a basis for our analyses.

## Climate reanalysis downscaling

ERA5-Land variables are statistically downscaled from 9-km to 100-m resolution on an hourly timescale. This process involves combining spatial interpolation methods with vertical gradients for each hourly timestep[11,70]. To downscale air temperature, we use monthly-hourly mean lapse rates derived from the station data described above. Due to the absence of spatially distributed measurements, no precipitation gradients were prescribed, keeping only the ERA5-Land native horizontal variability which shows larger precipitation amounts for higher elevation pixels (Supplementary Fig. S9). Wind speed was downscaled by leveraging the DEM of the catchment to identify areas that are either exposed to or sheltered from synoptic wind gradients[71,72]. Incoming shortwave and longwave radiation are downscaled using a vertical gradient approach found in the litterature[73]. Bias correction is performed monthly with the empirical quantile mapping (EQM) method, using all overlapping periods of station and reanalysis data to capture seasonal variability effectively. For precipitation, the bias correction is performed on daily sums, using the observed precipitation corrected for undercatch. Comparisons of downscaled meteorological variables against in-situ observations are shown in the Supplementary Material (Supplementary Figs. S4–7).

## Model set-up and initial conditions

Glacier outlines and debris-covered areas were based on the Randolph Glacier Inventory version 6.0[74,75] and manually adjusted where needed using the September 2022 Pléiades ortho-image[65,76]. Distributed debris thicknesses were derived from a relationship between remotely sensed land-surface temperature and in-situ measurements of debris thickness conducted on Kyzylsu Glacier[11]. Land cover was set up based on the PROBA-V[77] land cover and plant species (macro-types) were selected based on field observations and literature. The SOILGrids[78] soil product was used for soil thickness and soil composition. A visualization of the model spatial set-up is provided in the Supplementary Material (Supplementary Figs. S15, 16). Initial snow cover and snow albedo were based on the Landsat-7 scene acquired on 16 October 1999. Snow cover and snow surface albedo in shaded areas were set to the non-shaded area average of the corresponding elevation band. In the absence of snow depth information, we set the initial snow depth to 30 centimeters (Supplementary Fig. S18).

## Model parameters and evaluation

The model parameters were not calibrated through automatic calibration of multiple parameters against spatially integrated variables (e.g. discharge) but rather set from remote sensing data, in-situ observation, or literature. In our simulations, bare ice albedo is kept constant in time, but its spatial heterogeneity is accounted for by setting its value to the mean glacier albedo observed per 100-m elevation band for the 10 least snow-covered Landsat and Sentinel-2 scenes (Supplementary Fig. S17). While our model does not have a firn layer, using an altitudinally varying bare-ice albedo allows us to indirectly account for the fact that firn usually has a higher albedo than bare ice. The thermal conductivity and surface roughness of the supraglacial debris layer were optimized through energy balance simulations at the on-glacier AWS, where meteorological data and concurrent measurements of ablation were available[11,65]. Phenological parameters are the same as those used in Fugger et al.[11]. We tested three sets of parameters for the avalanche routine ($a = 0.10$ and $C = 145$, $a = 0.12$ and $C = 145$, $a = 0.14$ and $C = 145$) and selected the couple leading to the best agreement with remote-sensing observations of avalanche deposit locations ($a = 0.12$ and $C = 145$). We evaluated the model against measured surface albedo (Supplementary Fig. S24), glacier ablation stakes (Supplementary Fig. S25), glacier surface elevation change (Supplementary Fig. S26), observed snow depth (Supplementary Fig. S27), proglacial stream water level (Supplementary Fig. S28), remotely sensed snow cover (Supplementary Fig. S29–32) and glacier-wide geodetic mass balance for the period 2000–2019 (Fig. 5). We evaluate the gravitational snow redistribution against avalanche deposit outlines detected semi-automatically using all available Sentinel-1 scenes for the 2017–2023 period[79] and against avalanche deposits visible in the spatially distributed snow depth derived from the high-resolution Pléiades DEM differencing (Supplementary Figs. S21, 22). The performance of the model is summarized in Supplementary Table S2, and was satisfactory enough to not require further adjustment of the meteorological forcing or model parameters.

## Reporting summary

Further information on research design is available in the Nature Portfolio Reporting Summary linked to this article.

## Data availability

Model outputs, source data used to generate Figs. 2–8, as well as reconstructed meteorological time series and in-situ data collected at the study site which have been used for the model evaluation, are available at https://doi.org/10.5281/zenodo.14287505. The glacier outlines of [74] are available from the NSIDC via https://doi.org/10.7265/4m1f-gd79. The ALOS World 3D-30 (AW3D) DEM[69] was downloaded from http://www.opentopography.org. The Pléiades stereo-pairs were acquired within the scope of the CNES ISIS Programme. ERA5-Land reanalysis data[49,50] was downloaded from https://doi.org/10.24381/cds.e2161bac. MODIS daily snow cover product, Landsat-5/7/8/9 and Sentinel-2 scenes were downloaded using the Microsoft Planetary Computer https://planetarycomputer.microsoft.com/.

## Code availability

The source code of the land-surface model is available here: https://github.com/simonefatichi/TeC_Source_Code. The MATLAB code (version R2022b) required to analyze the model outputs and generate Figs. 2–8 paper is available at https://doi.org/10.5281/zenodo.14287505.

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

## Acknowledgements

This work was made possible with funding from the Swiss National Science Foundation (ASCENT Project 189890, Understanding snow, glacier and rivers response to climate in High Mountain Asia). It was also supported by the ERC Consolidator RAVEN project No. 772751, Rapid mass losses of debris-covered glaciers in High Mountain Asia. Fieldwork funding support for the repeated visits to Tajikistan was also received from the Swiss Polar Institute Flagship Programme PAMIR (SPI-FLAG-2021-001) and the Swiss National Science Foundation (HOPE Project 183633, High-elevation precipitation in High Mountain Asia). We would like to thank Firdavs Vosidov, Ubaydullo Ubaydulloev, Tojiddin Rasulzoda and Iskandarov Handullo from the Center for the Research of Glaciers, Tajik National Academy of Sciences (CRG-TAS), for their invaluable support over multiple field campaigns at the study site. We thank Nazrialo Sheralizoda, current director of CRG-TAS, and Tomas Saks from the University of Fribourg for their support in enabling and coordinating the ongoing collaborative monitoring and measurements at the site. Marin Kneib acknowledges the funding from the Swiss National Science Foundation (SNSF) under the Contribution of avalanches to glacier mass balance (CAIRN) Postdoc Mobility program (grant agreement P500PN_210739). We extend our thanks to Hamish Pritchard and Federico Covi at BAS for their help with the processing of lake water pressure data. Finally, we thank the photographer Jason Klimatsas for the photos he took which we use in Fig. 1b and Supplementary Fig. S1. Pleiades stereo imagery was acquired through the CNES ISIS programme.

## Author contributions

A.J. developed the concept of the study together with T.S, E.M, M.K., M.M. and F.P., who provided regular feedback and discussion for data collection, data analysis and model simulations. S.F. contributed to the set-up of the land-surface model and field data collection. P.B. designed the pluviometer station installed during the first field visit in 2021. A.K, H.N, A.H, K.K and F.H enabled and contributed to the in-situ data collection in Tajikistan. A.J. wrote the manuscript with input from T.S, E.M, M.K., S.F., P.B., M.M. and F.P.

## Competing interests

The authors declare no competing interests.
