## [Transparent Peer Review file · Communications Earth & Environment]

Snowfall decrease in recent years undermines glacier health and meltwater resources in the Northwestern Pamirs

Corresponding Author: Mr Achille Jouberton

Version 0:

Decision Letter:

Dear Mr Jouberton,

Your manuscript titled "Snowfall decrease in recent years undermines glacier health and meltwater resources in the Northwestern Pamirs" has now been seen by 2 reviewers, whose comments are appended below. In the light of their advice we regret to inform you that we cannot publish your manuscript in Communications Earth & Environment.

You will see that the reviewers raise substantive concerns regarding the robustness of analysis, arguments, and conclusions. Taking these points together with our editorial considerations, we are unable to conclude that your manuscript represents a sufficiently novel and compelling advance over the body of related work in the literature. Unfortunately, these reservations are sufficiently important to preclude publication of this study in Communications Earth & Environment.

Although we cannot offer to publish your manuscript, we believe the Editorial Board at Scientific Reports will find it interesting and recommend you transfer there. To transfer your manuscript there, please use our Link Redacted manuscript transfer portal. You will not have to re-supply manuscript metadata and files, unless you wish to make modifications. For more information, please see our [manuscript transfer FAQ](http://www.nature.com/authors/author_resources/transfer_manuscripts.html?WT.mc_id=EMI_NPG_1511_AUTHORTRANSF&WT.ec_id=AUTHOR) page.

We are sorry that we cannot be more positive on this occasion and thank you for the opportunity to consider your work.

Best regards,

Shin Sugiyama, PhD
Editorial Board Member
Communications Earth & Environment
orcid.org/0000-0001-5323-9558

Alireza Bahadori, PhD
Associate Editor
Communications Earth & Environment

Reviewers' comments:

Reviewer #1 (Remarks to the Author):

First and foremost, I would like to express my gratitude for submitting the manuscript titled "Snowfall decrease in recent years undermines glacier health and meltwater resources in the Northwestern Pamirs." The study focuses on the impact of recent snowfall reduction on glacier health and meltwater resources in the northwestern Pamir region, which holds scientific significance. However, after careful reading and deliberation, I recommend that this paper is not suitable for publication in the journal Communications Earth & Environment. The main reasons are as follows:

Insufficient Explanation of Results

- **Overly Simplified Explanation of Snowfall Reduction Causes:** The paper indicates a significant decrease in snowfall since 2018 but does not provide an in-depth analysis of the underlying climatic mechanisms. Although it mentions a decrease in total precipitation, it fails to explore in detail the regional climate characteristics and atmospheric circulation changes (such as shifts in the position of the westerlies, anomalies in regional moisture transport, etc.). Additionally, the explanation of the impact of snowfall reduction on glacier mass balance and meltwater resources is rather general, lacking analysis of the differential impacts on glaciers at different elevations and aspects. This makes the results less scientific and targeted.
- **Lack of Basis for Predicting Future Trends:** When discussing future changes in glaciers and water resources, the paper mainly speculates based on current observational data and model simulation results, without fully considering the uncertainties and complexities of future climate change. For instance, future temperature and precipitation changes may differ from the current observed trends, or human activities' impact on glaciers and water resources may intensify or mitigate. These factors could significantly affect the future changes in glaciers and water resources. Therefore, the current prediction results lack sufficient scientific basis and credibility, making it difficult to guide future water resource management and glacier conservation efforts.

Lack of Research Contribution and Innovation

- **Insufficient Contribution to Regional and Universal Research:** Although the snowfall changes in the northwestern Pamir region have certain regional uniqueness, the paper fails to fully explore and present this uniqueness. It also does not effectively compare the research results with glacier changes in other regions to reveal similarities and differences between regions. This makes the paper's contribution to universal research insufficient, as it cannot provide valuable references and insights for glacier research and water resource management in other areas.

Issues with Writing and Expression

- **Inaccurate and Unstandardized Language Expression:** Some sentences in the paper are inaccurately and standardly expressed, with issues such as awkward phrasing and ambiguity, affecting readers' understanding of the research content. For example, when describing model validation results, vague statements like "the model performs well" are used without specifying which indicators and standards led to this conclusion. Additionally, some professional terms and abbreviations are not clearly defined and explained, making it difficult for non-specialist readers to comprehend. In academic writing, accurate and standardized language expression is fundamental for ensuring the accurate transmission and communication of research information, yet the current writing quality needs improvement.

In summary, although the research topic of this paper has some scientific significance, due to the limitations in research methods, insufficient explanation of results, lack of research contribution and innovation, as well as issues with writing and expression, I recommend that this paper is not suitable for publication in the journal *Communications Earth & Environment*. I suggest that the authors further improve the research methods, conduct in-depth analysis of key factors, strengthen the explanation of results, enhance research innovation, and improve writing quality before considering submitting the paper to other more appropriate academic journals.

Wishing you success in your research endeavors!

Reviewer #2 (Remarks to the Author):

The study entitled 'Snowfall decrease in recent years undermines glacier health and meltwater resources in the Northwestern Pamirs' combines remote sensing data, in situ measurements and modelling to analyse changes in snow accumulation for the data-scarce Northwestern Pamir, where glacier melt water is critical for the water supply. Until recent years near to balanced glacier changes have been observed for this region despite the globally observed declining trend. Recent studies, however, suggest that the mass balance of glaciers in the Northwestern Pamir is also declining meanwhile. The spatial heterogeneity across the region, methodological limitations and an incomplete understanding of the drivers limit the current understanding. Among other drivers, increased precipitation is discussed to be a reason for the neutral mass balance observed in the past. Overall, the reasons for the mass balance behaviour in the region are poorly understood and in situ evidence is very limited. By combining in situ data, remote sensing information and modelling the authors contribute to the understanding of ongoing glacier mass and hydrological changes in a region.

The paper is well-written, and the structure is in general easy to follow. However, some elements in the result section are going beyond the presentation of results. I would suggest moving discussion statements within the results to the discussion section. As some parts of the discussion are rather general and not very much referring to the results of the study, the outcome of the analysis is rather underrepresented. They deserve more attention within the discussion. Adding a concluding remark to the main section paper (before the presentation of the methods) could further address this by highlighting the main results and their implication.

In addition, I have some specific comments:

L 88-91: The statement "Constraining snow amounts..." should be mentioned in the discussion/method section here as this is not a result.

L110-112: The statement is not very clear. Do you mean the thin snowpack is the reason for the melt-out or the warm spring conditions or even both? It should be clarified, what was observed and what is a hypothesis, here. Sometimes, as recent snowpack depletions in the Alps show, even a thick snowpack can melt out very fast.

L.151-54: The statement is not very clear, and a word seems to be missing at the end of the sentence. Is the albedo really the only reason? What's about an increase of sensitive heat fluxes, long-wave radiation or even changes in the firm? Please rephrase. And it would be helpful to underline the statement with a figure.

L166-169: Please move this statement to the discussion section.

L209-210: Do you mean water stored in glaciers? Please specify.

L242-249: Regarding the timing of the ambiguity of the snowfall decrease, it would be helpful to underline the choice of 2018 as a cut-off by some modelling results by referring to the mass balance and model forcing data evolution over time (Fig.5). How does the mass balance look like between 2012-2018 compared to the period before and after? What are the trends for the different periods?

L300-301: By 'their causes' do you mean the causes of the anomaly? Please rephrase this statement.

L323-324: Add some information on the precipitation seasonality and accumulation type of the glacier.

L429-431: Did the water stored in the firn change over the modelling period? Is there any indication of firn regime changes (warming of firn temperatures?).

L493ff: The model seems not to be validated against any in situ measurements in the accumulation area. Furthermore, no snow water equivalent measurements seem to be available for bias correction of precipitation measurements and/or reanalysis data. Are there any in situ measurements of snow accumulation available? As the paper addresses changes in snow accumulation and its effects on glacier mass balance, glacier mass balance changes in the accumulation zone should at least be discussed. A discussion could for instance be included in the section about the future evolution of the glacier under climate change. The uppermost zones of the glaciers are least affected by the temperature increase and future research should also include on those zones. I suggest to include this in the section starting at L269.

L298: This section appears rather quite generic. Try to better link it to the results presented. The title should be changed. Trajectories might not be the best word here. And I would rather mention the role of snowfall and glaciers for the water cycle than the cryosphere as the water supply is more a focus here than the cryosphere. If cryosphere is in the title I would expect other processes to be discussed including effects on permafrost etc.

Fig. 2: In 2022, the modelled snowpack is underestimated and in 2023 it is overestimated, what are the reasons? Can you please discuss this?

Fig. 4: The caption is missing some information. Specify in that these are model results. Also specify for which glacier is shown here. Add numbering (a/b) to the figure.

Fig. 5. Please improve the colours by using each colour only once. Also, there might possibly more intuitive colours to use (e.g. red colour tones for ablation, blue tones for accumulation). The caption also needs improvement. The figure shows mainly modelled mass fluxes; this should be specified. In the text, it should be discussed why some mass fluxes are not shown/resolved in the model (I think Ice melt stands for melt of glacier ice and firn. Sublimation seems to be negative only, is there no deposition occurring resp. resolved in the model resolution? And what's about the refreezing melt water in the snowpack (as described in L429ff)). Furthermore, the caption could provide a bit more guidance to the reader (Information where on the figure the different points mentioned can be found could be helpful such as "... rectangles on the top", "geodetic mass balance purple line...").

Fig. 6. For b and c. use white background for text. Put b) and c) to the top left corner similar to a). Add a scale to the maps.

Fig. 7 Numbering (a)/b/...) is missing in the figure. I would also suggest to use the same colours for each mass flux as in Figure 5.

Fig. 8 Numbering (a)/b)) is missing in the figure.

Version 1:

Decision Letter:

Dear Mr Jouberton,

Your revised manuscript titled "Snowfall decrease in recent years undermines glacier health and meltwater resources in the Northwestern Pamirs" has now been seen by our original reviewer 1 and new reviewers 3 and 4 who replace the original reviewer 2 (reviewer 2 was not available for further comments). All comments appear below. In light of their advice we are delighted to say that we are happy, in principle, to publish a suitably revised version in Communications Earth & Environment.

We therefore invite you to revise your paper one last time to address the remaining concerns of our reviewers 3 and 4. At the same time we ask that you edit your manuscript to comply with our format requirements and to maximise the accessibility and therefore the impact of your work.

EDITORIAL REQUESTS:

*****Please take care to match our formatting and policy requirements. We will check revised manuscript and return manuscripts that do not comply. Such requests will lead to delays. *****

SUBMISSION INFORMATION:

OPEN ACCESS:

Communications Earth & Environment is a fully open access journal. Articles are made freely accessible on publication. For further information about article processing charges, open access funding, and advice and support from Nature Portfolio, please visit <https://www.nature.com/commsenv/open-access>

Link Redacted

Best regards,

Shin Sugiyama, PhD
Editorial Board Member
Communications Earth & Environment
orcid.org/0000-0001-5323-9558

Alireza Bahadori, PhD
Associate Editor
Communications Earth & Environment
Consulting Editor
Communications Sustainability

REVIEWERS' COMMENTS:

Reviewer #1 (Remarks to the Author):

Agree to publish

Reviewer #3 (Remarks to the Author):

Please see attachments

Reviewer #4 (Remarks to the Author):

Please see the attachment file if texts here are not in line.

Review of the manuscript "Snowfall decrease in recent years undermines glacier health and meltwater resources in the Northwestern Pamirs" written by Jouberton et al.

The manuscript has already been reviewed by two other reviewers before me and therefore the manuscript is in very good shape. After reading the comments of reviewer#1 and response from the authors, I feel the authors have attempted to

respond well with as much as possible explanations (from the literatures, for such as: reasons/drivers for reduced snowfall in recent period and futuristic perspective of the glaciers/snowfall in the region) and additional analysis. I also agree with the authors that some of the explanations reviewer#1 asked for (e.g., reasons/drivers for reduced snowfall and futuristic perspective of the glaciers/snowfall) are difficult to point out based on the study objectives/tasks the authors set and in the literature such large-scale change understanding has not yet been investigated/understood well. Therefore, in my opinions, the explanations and modifications by the authors are good and sufficient considering their focus of the study, which is exploring the glacier changes in the region using modelling and possible explanations of the drivers and conditions.

Below I point out some of the small changes, majorly stylistic and language related, that needs to be fixed before its acceptance.

Abstract

1. I think opening sentence, L 22-24 (in the marked_up manuscript file), is a bit too long and the last part of it is not well paired with the first part of it. What I try to mean is, the first part reads ‘..world’s last relatively healthy mountain glaciers..’ which sounds positive, however, the last part reads ‘..causes of this anomalous glacier state are not known.’ Which sounds negative but the different behaviour of glaciers here are not presented before, so it sounds like where does the ‘anomalous glacier state’ come from. I would recommend rephrasing the sentence and dividing into two parts maybe for better read.
2. In L 27-29, snowfall should be placed before snow depth which is more logical.
3. L 29-31, does the authors want to rephrase it to ‘..cause of higher/increased mass losses..’ or just ‘mass losses’ is fine? – because I see that the recent years mass losses are higher than the decade let’s say 2000s (which the authors already mention in the next line, L 31-32).

Introduction

1. L 48-51, here the authors says that their modelling period is 1999-2023 (also in L64 later), but in abstract it was mentioned as 2000-2023. Please check. Additionally, I would also quickly point out the study catchment name here, may be in this way ‘..in Tajikstan (Kyzylsu catchment),..’ because until reading, the readers will already be curious to know about the site. Moreover, the appearance of the site name in the next para is bit sudden. Therefore, I feel mentioning quickly in L48-51 would be a good choice.

(before going through the results/discussion, I was more curious to learn about the methods, therefore, I present my opinions on Methods before Results)

Methods

1. L 328-329, here I think ‘while’ is not necessary and reads stylistically awkward. Instead, ‘therefore’ may be used.

Results

1. L 93-95, I think this sentence belongs to the Methods section, somewhere. It sounds a bit sudden to appear with less context for it. Please check if the authors can place it somewhere in the Methods section. The hydrological year has already been mentioned in the ‘land-surface model’ sub-section in Methods below.

** Visit Nature Portfolio's author and referees' website at www.nature.com/authors for information about policies, services and author benefits**

REVISION STATEMENT - Response to reviewers

Snowfall decrease in recent years undermines glacier health and meltwater resources in the Northwestern Pamirs.

by

Achille Jouberton, Thomas E. Shaw, Evan S Miles, Marin Kneib, Stefan Fugger, Pascal Buri, Michael McCarthy, Abdulhamid Kayumov, Hofiz Navruzshoev, Ardamehr Halimov, Khusrav Kabutov, Farrukh Homidov, Francesca Pellicciotti.

GENERAL REVISION

We would like to thank you for reconsidering our manuscript entitled “Snowfall decrease in recent years undermines glacier health and meltwater resources in the Northwestern Pamirs” for publication in *Communications Earth & Environment*, which we revised after receiving this first round of reviews.

The main issues raised by the reviewers and editor were: i) lack of explanations of the decrease in snowfall that we report, ii) lack of clarity related to some statements in our result sections, iii) lack of reference to our results in the discussion sections and iv) concerns regarding model validation.

In response to these comments:

- We added a paragraph in the Discussion section “*Snow cover and precipitation decline across the Northwestern Pamirs*” that uses the current literature review to frame the need for future analysis on the causes of this precipitation decline. We feel this is an important addition to the paper and a great topic of future research that we hope the climate community will pick upon.
- We conducted new analyses to clarify the role of snowfall decline on snowpack melt-out timing and glacier mass loss, which consisted of linear regression analyses and investigations into changes in energy fluxes. The results were added as text in the main manuscript and as 3 new figures in the Supporting Information (SI).
- We now refer more to our results in the discussion sections, which contain less generic statements. As a result, the discussion sub-section “*Future trajectories of the Pamir glaciers*” changed substantially.
- We modified the terminology used in the manuscript to make more explicit our use of snow water equivalent measurements for the model calibration, more specifically in the bias correction of ERA5-Land precipitation. We now indicate early on in the manuscript the presence of Table S2 in the SI which summarizes the performance of the model against ground and remote-sensing observations. We took this opportunity to improve the structure of the Supporting Information, facilitating access to the demonstration of the model performance, in addition to the model comparison

against observations which is included in Figures 2, 3 and 5 of the main manuscript.

We would like to thank the reviewers for their constructive comments, which further contributed to improving the manuscript. We very much hope that the revised manuscript is now appropriate for publication in *Communications Earth & Environment*.

REVIEWER #1:

General comments:

First and foremost, I would like to express my gratitude for submitting the manuscript titled "Snowfall decrease in recent years undermines glacier health and meltwater resources in the Northwestern Pamirs." The study focuses on the impact of recent snowfall reduction on glacier health and meltwater resources in the northwestern Pamir region, which holds scientific significance. However, after careful reading and deliberation, I recommend that this paper is not suitable for publication in the journal *Communications Earth & Environment*.

We thank Reviewer 1 for their comment about the drivers of precipitation decline, which we find important. We are surprised however that Reviewer 1 does not see the fundamental advances of our study and recommends its rejection. We disagree with several points raised by the reviewer, which we address below. We have, however, tried to receive the criticisms constructively and have adapted our manuscript in response to the concerns or misunderstandings raised by the reviewer. Line or figure numbers refer to the revised manuscript unless stated otherwise.

Major comments:

Insufficient Explanation of Results

- **Overly Simplified Explanation of Snowfall Reduction Causes:** The paper indicates a significant decrease in snowfall since 2018 but does not provide an in-depth analysis of the underlying climatic mechanisms. Although it mentions a decrease in total precipitation, it fails to explore in detail the regional climate characteristics and atmospheric circulation changes (such as shifts in the position of the westerlies, anomalies in regional moisture transport, etc.). Additionally, the explanation of the impact of snowfall reduction on glacier mass balance and meltwater resources is rather general, lacking analysis of the differential impacts on glaciers at different elevations and aspects. This makes the results less scientific and targeted.

We thank the reviewer for suggesting to include additional explanations of the decrease in snowfall that we obtain. This is a useful suggestion - but we politely but strongly disagree with the reviewer's statement that we insufficiently explain our

results and that this makes them “less scientific”. Our study investigates drivers in glacier and snow changes since 2000 using a sophisticated model that accounts for all components of the mass balance, and identifies a snowfall reduction since 2018 as a key driver of recent glacier mass loss. This is a novel, fundamental result, given the important current scientific debate on the state of Pamir glaciers (neutral or slightly negative mass balances) and their future trajectory. It is particularly so given the recent suggestion from remote sensing studies of a possible shift in mass balance regime. We did not explore in detail the reasons for this snowfall (and total precipitation) decline, as they would require **a study per se on westerly strength and moisture sources**, as Reviewer 1 mentions, within the domain of atmospheric dynamics and climate modelling expertise and with tools from those disciplines. While this is outside of the scope of our study, this is nevertheless a good point raised by the reviewer, and we have made an effort to investigate possible reasons in the existing literature. To the best of our knowledge, **there is no explanation available (yet) from the atmospheric and climate community on the decrease in precipitation since 2018**. We added a short paragraph in the Discussion Section that uses the current literature review that this comment prompted (and that we discuss below) to frame the need for future analysis on the causes of this decline. We feel this is an important addition to the paper and a great topic of future research that we hope the climate community will pick upon.

The northwestern Pamirs receive most of their annual precipitation during the winter and spring months (Chen et al. 2025), and generally receive moisture from the mid-latitude westerlies (Yan et al. 2019). The understanding of western disturbances (westerlies) has seen great advances in the last decade due to the development of analysis techniques such as tracking algorithms and the increasing availability of high-resolution weather and climate models, which are the focus of the recent comprehensive review of Hunt et al. (2025). The contribution of westerlies to winter precipitation in the western Himalayas and surrounding regions is still unknown but is likely above 50% (Hunt et al. 2025). Over the last 70 years, western disturbances have increased significantly and this was attributed to a strengthening of the subtropical jet (Hunt et al. 2024). Not all studies agree on this, however, and there is no agreement on trends of winter precipitation in the region (Hunt et al., 2025). which are a complex function of period and location, with measurements indicating strong decadal variability. The drivers of decadal variability in western disturbances and precipitation in the Pamirs are not fully understood but could be linked to different phases of the Pacific Decadal Oscillation (Jiang et al. 2021, Jiang & Zhou 2023), El Niño-Southern Oscillation and North Atlantic sea surface temperature anomalies (Yao et al. 2024), and deserve further investigation. Long-term precipitation measurements in Tajikistan are limited, and existing studies do not include data from 2019 onward (e.g. Gulahmadov et al. 2023). We have added some of these points in the discussion section. (Lines 245-255).

Finally, the Reviewer mentions that our explanations are “lacking analysis of the

differential impacts on glaciers at different elevations”. We do not agree with their comment, as we are providing the simulated glacier surface mass balance **per elevation bands** in Figure 6a, and the changes in runoff components (including icemelt) per elevation bands in Figure 7. Specifically, we find that the snowfall decline was the largest at glacierized elevations (Fig. 4), which resulted in little net mass gain in the accumulation area (above 4200 m a.s.l., Fig 6a), increasing icemelt above 3500 m a.s.l. and declining snowmelt at all elevations (Fig. 7c).

- **Lack of Basis for Predicting Future Trends:** When discussing future changes in glaciers and water resources, the paper mainly speculates based on current observational data and model simulation results, without fully considering the uncertainties and complexities of future climate change. For instance, future temperature and precipitation changes may differ from the current observed trends, or human activities' impact on glaciers and water resources may intensify or mitigate. These factors could significantly affect the future changes in glaciers and water resources. Therefore, the current prediction results lack sufficient scientific basis and credibility, making it difficult to guide future water resource management and glacier conservation efforts.

We are surprised by this comment since we **do not make any future predictions** of glacier and water resource changes in our manuscript, do not claim so and state very clearly that our study period is **1999 to 2023** (L50 and L64 of the original manuscript). We only provide a discussion on the "Future trajectories of the Pamir cryosphere" in the Discussion section, where we briefly mention that the future of Pamir glaciers is the focus of intense debate and research, due to their anomalous behaviour in recent decades (Farinotti et al. 2020, De Kok et al. 2020). We then discuss that substantial changes in snowfall fraction might be expected when looking at projected temperature changes into the future, and that the impact on glacier mass balance should be investigated (line 310 of the original manuscript). Importantly, the physical basis of the model, combined with a very careful adaptation of the model inputs, *does* provide the toolset to explore future changes for snow and glacier mass balance. And while this is not the focus of our work, we found it important to discuss the future perspective that our new findings open and now enables.

Lack of Research Contribution and Innovation

- **Insufficient Contribution to Regional and Universal Research:** Although the snowfall changes in the northwestern Pamir region have certain regional uniqueness, the paper fails to fully explore and present this uniqueness. It also does not effectively compare the research results with glacier changes in other regions to reveal similarities and differences between regions. This makes the paper's contribution to universal research insufficient, as it cannot provide valuable references and insights for glacier research and water resource management in other areas.

The reviewer argues that our study fails to explore the uniqueness of the Pamir region and the relevance of our results for other glaciers in the world. Whereas we, in fact, do *exactly that*: we explore with physically-based tools the unique behaviour of the region and provide physical explanations, backed by ground observations, for that unique behaviour, its recent changes and their main drivers. In the introduction of the manuscript we highlight some of the key unique features of the Pamir region: glaciers showing limited mass loss (L22-24 and L40-42 of the original manuscript), strong reliance of downstream areas to mountain cryosphere due to low precipitation in the lowlands (L37-39), and disruption of monitoring network following the collapse of the Soviet Union (L53-54).

We are not sure what the reviewer means here with “Universal” research. We want to stress, however, that our study is **a very substantial contribution to regional and global research**, with wide implications for the understanding of glacier response to regional climatic variability. The Pamirs are one of the few/only mountain regions in the world where glacier mass balance has not been substantially negative over the past decades. Here we show that this has changed very recently, and provide the first mechanistic explanation for this inflection point in the glacier mass balance regime. This is an extremely important update to the current state of knowledge on the cryosphere in the Pamirs, but also to the current state of knowledge on the global cryosphere as a whole. Our study is also of global relevance in disentangling the importance of precipitation in moderating short- to mid-term glacier response to climate change: precipitation signals can overwhelm temperature trends over decadal scales, something that can take place across climates and continents (Hugonnet et al., 2021), and our study demonstrates one such instance with unprecedented physical explanations: we identify precipitation changes as the major driver for a *rapid* change in glacier mass balance (and possibly a temporary one part of larger fluctuations), and open up the key question of whether precipitation might return to previous levels and how much precipitation would be able to 'negate' the detrimental temperature increases. A second key result of global relevance is the ability of the cryosphere to buffer those short to medium-term precipitation deficits. This is also in the focus of current research on other continents, for example in Chile where a megadrought which started in 2010 due to precipitation deficit has both put stress on and increased the reliance on the cryosphere (McCarthy et al. 2022).

The reviewer states that we do not compare our results with glacier changes in other regions. We feel that a comparison with glacier changes in other regions of the world would make little sense, as this is not a review paper nor a global modelling study, and the uniqueness of the region is actually the focus of our work. It would also be very difficult to decide what to compare our results to. The only global estimate of recent glacier changes (Hugonnet et al., 2021, presenting an assessment of glacier mass balance from space), is indeed our starting point, as the authors of that study suggested the first evidence of a potential shift in glacier mass balance regime.

Global remote sensing studies are great at providing a consistent global picture, but cannot provide any cause or mechanism for those patterns. Besides this remote sensing study, there is a plethora of mass balance studies at various scales, but differences in study periods, methods (modelling and remote-sensing, and assumptions within each category) used to estimate these changes across the globe would make the comparison almost impossible. Our methodology to assess changes in high-elevation water resources, which combines in-situ data, multi-source remote sensing observations and a detailed land-surface model, allows us to identify the mechanics of changes in snow and glacier resources. It could therefore be applied more broadly, for example to the Pamir-Karakoram region to finally resolve the competing hypotheses that have been suggested to explain the recent anomalous glacier state (Farinotti et al. 2020).

Issues with Writing and Expression

- Inaccurate and Unstandardized Language Expression: Some sentences in the paper are inaccurately and standardly expressed, with issues such as awkward phrasing and ambiguity, affecting readers' understanding of the research content. For example, when describing model validation results, vague statements like "the model performs well" are used without specifying which indicators and standards led to this conclusion. Additionally, some professional terms and abbreviations are not clearly defined and explained, making it difficult for non-specialist readers to comprehend. In academic writing, accurate and standardized language expression is fundamental for ensuring the accurate transmission and communication of research information, yet the current writing quality needs improvement.

We disagree with the Reviewer's assessment of our use of ambiguous language, and it is difficult to understand what the reviewer refers to since they do not provide any example of sentences that are "inaccurately and standardly expressed" or of "awkward phrasing". We also note that the second reviewer found instead the paper to be well written. In response to the reviewer's point that we use "vague statements" to describe the model validation, we would like to point out that nowhere in the original manuscript did we use the statement "the model performs well". The closest statement would be Lines 500-502 (original manuscript): "*The performance of the model is summarized in Table S2, and was satisfactory enough to not require further adjustment of the meteorological forcing or model parameters*", with Table S2 providing all of the metrics or 'indicators' that the reviewer says are missing. We now refer to Table S2 also in the introduction (Line 65), to make sure that the reader also looks at the Supporting Information if interested or concerned by the model evaluation.

Regardless, we have re-read the manuscript and brought improvements to the language where possible.

REVIEWER #2:

General comments:

The study entitled ‘Snowfall decrease in recent years undermines glacier health and meltwater resources in the Northwestern Pamirs’ combines remote sensing data, in situ measurements and modelling to analyse changes in snow accumulation for the data-scarce Northwestern Pamir, where glacier melt water is critical for the water supply. Until recent years near to balanced glacier changes have been observed for this region despite the globally observed declining trend. Recent studies, however, suggest that the mass balance of glaciers in the Northwestern Pamir is also declining meanwhile. The spatial heterogeneity across the region, methodological limitations and an incomplete understanding of the drivers limit the current understanding. Among other drivers, increased precipitation is discussed to be a reason for the neutral mass balance observed in the past. Overall, the reasons for the mass balance behaviour in the region are poorly understood and in situ evidence is very limited. By combining in situ data, remote sensing information and modelling the authors contribute to the understanding of ongoing glacier mass and hydrological changes in a region.

The paper is well-written, and the structure is in general easy to follow. However, some elements in the result section are going beyond the presentation of results. I would suggest moving discussion statements within the results to the discussion section. As some parts of the discussion are rather general and not very much referring to the results of the study, the outcome of the analysis is rather underrepresented. They deserve more attention within the discussion. Adding a concluding remark to the main section paper (before the presentation of the methods) could further address this by highlighting the main results and their implication.

We thank the reviewer for their detailed and constructive comments which we believe have improved the quality and clarity of the manuscript. We respond to the specific comments below. Line or figure numbers refer to the revised manuscript unless otherwise stated.

Specific comments:

L 88-91: The statement “Constraining snow amounts...” should be mentioned in the discussion/method section here as this is not a result.

Thank you. We decided to remove this statement altogether, as similar statements are already present at the end of the introduction (Line 61) and in the discussion section (Lines 222-226, 278-281).

L110-112: The statement is not very clear. Do you mean the thin snowpack is the reason for the melt-out or the warm spring conditions or even both? It should be clarified, what was observed and what is a hypothesis, here. Sometimes, as recent snowpack depletions in the Alps show, even a thick snowpack can melt out very fast.

Thank you for this comment, which is a very useful one. We agree that a thin snowpack is not necessarily the only reason explaining earlier snowpack melt-out, and that warm spring air temperatures could also cause a rapid melt-out. We now investigated these drivers by conducting linear regressions between the yearly mean snowpack melt-out date and the yearly i) mean snowpack height on March 1st (date close to peak SWE) ii) March-May mean air temperature, iii) November-February snowfall, and iv) March-May snowfall. We find the highest correlation with the snow depth on March 1st ($R^2 = 0.60$), followed by the Nov-Feb snowfall ($R^2 = 0.54$), and then by the spring air temperature ($R^2 = 0.43$). This shows that the snowpack's earlier melt-out from 2018 onwards is mostly controlled by the lower winter snowfall and snow height, but this is enhanced by slightly warmer spring air temperatures (+0.21°C between 2000-2018 and 2018-2023).

We have added the linear regression and corresponding scatterplots in a new Figure S41, and reformulated the corresponding text in the manuscript (Lines 108-111), which now reads as follow: “*Correlations between the simulated snowpack melt-out date and the reconstructed climatic forcing (Fig. S41) suggest that earlier seasonal snowpack depletion was mainly caused by thinner winter snowpacks, and secondly by warmer spring conditions (+0.21°C, Fig. S36).*”

L.151-54: The statement is not very clear, and a word seems to be missing at the end of the sentence. Is the albedo really the only reason? What's about an increase of sensitive heat fluxes, long-wave radiation or even changes in the firn? Please rephrase. And it would be helpful to underline the statement with a figure.

We thank the reviewer for this comment. Following their suggestion, we looked at changes between 2000-2018 and 2018-2023 for the following variables: sensible heat flux (+0.6 W/m²), incoming long-wave radiation (-2.1 W/m²), incoming shortwave radiation (+1.5 W/m²), net radiation (+3.7 W/m²), latent heat (+0.3 W/m²), and surface albedo (-0.023). As can be seen from these numbers, the increase in incoming shortwave radiation and in sensible heat flux is compensated by the decrease in incoming longwave radiation. Using the whole period averaged incoming shortwave radiation of 165 W/m², the decrease in surface albedo corresponds to a 3.8 W/m² increase in net shortwave radiation, which explains most of the net radiation increase and is substantially higher than the increases in sensible and latent heat fluxes. We provide a figure below and in the SI (Fig. R1/S44) with the energy fluxes and albedo comparison. Attributing the decrease in glacier mass balance (in mm w.e.) to these different processes and changes in energy fluxes is

not straightforward due to their interdependence and non-linear effects on the mass balance.

We have reformulated the statement in an attempt to make it clearer (Lines 145-151). It now reads as follows: “The snowfall reduction contributes to a substantial part (-0.31 m w.e./yr) of the additional glacier mass loss (-0.71 m w.e./yr), which was also due to enhanced ice melt (+0.32 m w.e./yr). However, the snowfall decline contribution to this additional mass loss could be higher due to its role in the decrease of snow cover and surface albedo (-0.023, Fig. S44), which in turns contributes to higher ice melt, and which is difficult to quantify due to the concurrent effect of warmer summer temperatures (+1.1 °C, Fig. S36) on snow cover duration, albedo and ice melt rates.”

Additionally, we conducted a linear regression analysis between simulated glacier mass balance, snowfall and summer air temperatures. The results are mentioned in the revised manuscript (Lines 151-152) and shown in a new figure in the SI (Fig. S42).

Figure R1/S44. Energy fluxes and surface albedo of Kyzylsu Glacier, for the periods 2000-2018 and 2018-2023 (top panel), and their differences (bottom panel). These fluxes were averaged over the whole area of Kyzylsu Glacier. The variables are, from left to right: SWin = incoming shortwave radiation, LWin = incoming longwave radiation, H = sensible heat flux, L = latent heat flux, Rn = net radiation, Albedo = surface albedo.

L166-169: Please move this statement to the discussion section.

We moved the statement to the discussion (Lines 304-307).

L209-210: Do you mean water stored in glaciers? Please specify.

We mean water stored in snow and glaciers in the solid state, from which originates snowmelt and icemelt. We have specified by adding “frozen” in front of “water storages” (Lines 208-209).

L242-249: Regarding the timing of the ambiguity of the snowfall decrease, it would be helpful to underline the choice of 2018 as a cut-off by some modelling results by referring to the mass balance and model forcing data evolution over time (Fig.5). How does the mass balance look like between 2012-2018 compared to the period before and after? What are the trends for the different periods?

The simulated glacier mass balance for the period 2000-2012 is -0.06 m w.e./yr, while it is -0.22 ± 0.27 m w.e./yr for the period 2012-2018 and -0.82 m w.e./yr for 2018-2023. This shows that the mass loss has been limited before 2012, moderate between 2012 and 2018 and high from 2018 onwards, which explains our choice of 2018 for the cut-off.

We are not sure we understand the suggestion or comment of the reviewer, as we believe we had addressed this issue in the original manuscript. We mentioned in the original manuscript that “We selected the end of the hydrological year 2018 (September 30th) as a middle point for our two sub-periods of analysis based on the largest changes in mean annual precipitation and glacier mass balance (Fig. S34), yet snowfall amounts have generally been below average since 2012 (Fig. 2).” (L242-246 of the original manuscript). We have now added the numbers (given above) in the caption of Fig. S34.

L300-301: By ‘their causes’ do you mean the causes of the anomaly? Please rephrase this statement.

Yes, we meant the causes of the anomaly. However, we have removed this statement since it is rather general and does not directly refer to the results of the study, which was one of the overarching comments of Reviewer #2. The mentions of the anomalous glacier state and its causes are already provided in the introduction (Lines 41-42).

L323-324: Add some information on the precipitation seasonality and accumulation type of the glacier.

We have now added this information following the reviewer's suggestion (Lines 328-329). The text now reads: *"Most of the precipitation falls in winter and spring (Fugger et al. 2024), while the glaciers are considered as winter-accumulation type (Huang et al. 2022)."*

L429-431: Did the water stored in the firn change over the modelling period? Is there any indication of firn regime changes (warming of firn temperatures?).

This is a useful comment. We are not able to quantify changes in water stored in the firn or changes in firn temperature because the model does not include a firn layer. Using a more detailed snowpack model with a representation of firn would have been hampered by the absence of ground observation in the accumulation area to validate the model results (cf. the answer to the reviewer's next comment below). Long-term firn modeling was conducted by Kronenberg et al. (2022) at Abramov Glacier, which is located only 60 kilometers north of Kyzylsu Glacier, for the period 1968-2020. They found that periods of negative mass balance led to a loss of firn pore space and a decrease in internal accumulation (refreezing and storage of water in the firn). They report decadal internal accumulation ranging from 0.08 m w.e. a⁻¹ in 2011-2020 to 0.13 m w.e. a⁻¹ in 1988-1998. The decadal differences in internal accumulation (+/- 0.05 m w.e. a⁻¹) are relatively small compared to our simulated changes in mass inputs at Kyzylsu Glacier due to snowfall (-0.31 m w.e./yr) and avalanche changes (-0.13 m w.e./yr). While changes in firn regimes might have occurred at Kyzylsu Glacier in the last two decades, this is likely to be a minor process compared to precipitation changes. We have now included these considerations in the Supporting Information (SI, Section 4.5), and mention it briefly in the main manuscript (Lines 437-439). We agree however that it would deserve further research.

L493ff: The model seems not to be validated against any in situ measurements in the accumulation area. Furthermore, no snow water equivalent measurements seem to be available for bias correction of precipitation measurements and/or reanalysis data. Are there any in situ measurements of snow accumulation available? As the paper addresses changes in snow accumulation and its effects on glacier mass balance, glacier mass balance changes in the accumulation zone should at least be discussed. A discussion could for instance be included in the section about the future evolution of the glacier under climate change. The uppermost zones of the glaciers are least affected by the temperature increase and future research should also include on those zones. I suggest to include this in the section starting at L269.

We measured snow water equivalent using lake pressure transducers installed over the winter 2021-2022 near the Pluviometer station, following the approach of Pritchard et al. (2021). This data was used to further adjust the undercatch correction of the pluviometer time series, which was then used for the bias correction of the reanalysis. We thank the reviewer for pointing out this lack of clarity and have now made this more explicit in the manuscript (Lines 63, 279-280, 362).

The reviewer is correct that the model was not evaluated against in-situ measurements in the accumulation area of the glacier, as, as it is common for most glaciers and studies, this is not accessible on foot (icefall and avalanche danger) and the usage of helicopters is very challenging in Tajikistan. Instead, we use a combination of in-situ measurements in the ablation area and remote sensing products covering the **whole** glacier to evaluate the model (thus including also the accumulation area), which we mention in Lines 273-276, and assume based on this that the model is performing relatively well in the accumulation area too. As suggested, we have added two sentences in the discussion sections, referring to our results in accumulation areas (Lines 302-205).

L298: This section appears rather quite generic. Try to better link it to the results presented. The title should be changed. Trajectories might not be the best word here. And I would rather mention the role of snowfall and glaciers for the water cycle than the cryosphere as the water supply is more a focus here than the cryosphere. If cryosphere is in the title I would expect other processes to be discussed including effects on permafrost etc.

We have modified the title, replacing “cryosphere” with “glaciers”, since it is true that the glaciers are the main focus of the discussion section. We have modified the section substantially, removing overly generic sentences and referring more to our results. We thank the reviewer for this constructive comment.

Fig. 2: In 2022, the modelled snowpack is underestimated and in 2023 it is overestimated, what are the reasons? Can you please discuss this?

Thank you for this detailed question, which we also asked ourselves when performing the model evaluation against snow depth. While we could not pinpoint the exact reasons, we hypothesise that the slight differences in modelled snowpack performance could be due to i) the meteorological forcing, even if ERA5-Land is bias-corrected, it might not fully represent the difference in meteorological conditions which occurred during these two winters ii) wind redistribution occurring during snowfall events, which is not represented in the model and could induce slight changes in the snow depth. We note that the performance of the snow depth simulations is still very good (mean bias of 0.029 m, R^2 of 0.92, cf. Fig. S27), such that we do not feel the need to discuss the remaining discrepancies in the main manuscript, but have added these discussion points in the caption of Fig. S27.

Fig. 4: The caption is missing some information. Specify in that these are model results. Also specify for which glacier is shown here. Add numbering (a/b) to the figure.

We have added the missing information and the numbering, thank you for pointing this out.

Fig. 5. Please improve the colours by using each colour only once. Also, there might possibly more intuitive colours to use (e.g. red colour tones for ablation, blue tones for accumulation). The caption also needs improvement. The figure shows mainly modelled mass fluxes; this should be specified. In the text, it should be discussed why some mass fluxes are not shown/resolved in the model (I think Ice melt stands for melt of glacier ice and firn. Sublimation seems to be negative only, is there no deposition occurring resp. resolved in the model resolution? And what's about the refreezing melt water in the snowpack (as described in L429ff)). Furthermore, the caption could provide a bit more guidance to the reader (Information where on the figure the different points mentioned can be found could be helpful such as "... rectangles on the top", "geodetic mass balance purple line...").

Following this comment, and to be consistent with Figure 7, we now show explicitly sublimation and evapotranspiration (ET) as two separate mass fluxes. Sublimation and deposition are resolved by the model, such that positive mass flux due to deposition is possible, but when aggregated at the yearly and glacier scales, the net sublimation and ET mass fluxes are negative (mass lost to the atmosphere).

We do not show refreezing meltwater in the snowpack in this figure, as meltwater refreezing does not change the mass balance of the snowpack nor of the glacier. Refreezing is accounted for in the "snowmelt" variable shown as the amount of snowmelt computed at each timestep based on the energy balance of the snowpack, which considers the energy consumed by the melting of liquid water refrozen contained in the snowpack. This liquid water either originated from snowmelt or rainfall that infiltrated the snowpack. In the latter case, rainfall may contribute to the glacier mass balance, but this effect is likely to be minimal.

We have modified the legend and the colors following the suggestions of the reviewer, using blue tones for accumulation, but keeping the original colors for ablation to be consistent with the colours of mass fluxes used in Figure 7.

Fig. 6. For b and c. use white background for text. Put b) and c) to the top left corner similar to a). Add a scale to the maps.

We have now implemented these suggestions.

Fig. 7 Numbering (a)/b/..)) is missing in the figure. I would also suggest to use the same colours for each mass flux as in Figure 5.

We have added the letters; thank you for pointing this out. Regarding the colours, please see our response to your comment on Figure 5.

Fig. 8 Numbering (a)/b)) is missing in the figure.

We have added the letters; thank you for pointing this out.

References used in this response to reviewers

Chen, S., Rugenstein, J. K. C., & Mulch, A. (2025). Stable isotope composition of surface waters across the Pamir, Central Asia: Implications of precipitation seasonality. *Journal of Hydrology*, 653, 132815. <https://doi.org/10.1016/J.JHYDROL.2025.132815>

Farinotti, D., Immerzeel, W. W., de Kok, R. J., Quincey, D. J., & Dehecq, A. (2020). Manifestations and mechanisms of the Karakoram glacier Anomaly. *Nature Geoscience* 2020 13:1, 13(1), 8–16. <https://doi.org/10.1038/s41561-019-0513-5>

Gulahmadov, N., Chen, Y., Gulakhmadov, M., Satti, Z., Naveed, M., Davlyatov, R., Ali, S., & Gulakhmadov, A. (2023). Assessment of Temperature, Precipitation, and Snow Cover at Different Altitudes of the Varzob River Basin in Tajikistan. *Applied Sciences* 2023, Vol. 13, Page 5583, 13(9), 5583. <https://doi.org/10.3390/APP13095583>

Hugonnet, R., McNabb, R., Berthier, E., Menounos, B., Nuth, C., Girod, L., Farinotti, D., Huss, M., Dussailant, I., Brun, F., & Käab, A. (2021). Accelerated global glacier mass loss in the early twenty-first century. *Nature*, 592(7856), 726–731. <https://doi.org/10.1038/s41586-021-03436-z>

Hunt, K. M. R. (2024). Increasing frequency and lengthening season of western disturbances are linked to increasing strength and delayed northward migration of the subtropical jet. *Weather and Climate Dynamics*, 5(1), 345–356. <https://doi.org/10.5194/WCD-5-345-2024>

Hunt, K. M. R., Baudouin, J.-P., Turner, A. G., Dimri, A. P., Jeelani, G., Chattopadhyay, R., Cannon, F., Arulalan, T., Shekhar, M. S., Sabin, T. P., & Palazzi, E. (2025). Western disturbances and climate variability: a review of recent

developments. *Weather and Climate Dynamics*, 6(1), 43–112. <https://doi.org/10.5194/wcd-6-43-2025>

Jiang, J., Zhou, T., Chen, X., & Wu, B. (2021). Central Asian Precipitation Shaped by the Tropical Pacific Decadal Variability and the Atlantic Multidecadal Variability. *Journal of Climate*, 34(18), 7541–7553. <https://doi.org/10.1175/JCLI-D-20-0905.1>

Jiang, J., & Zhou, T. (2023). Agricultural drought over water-scarce Central Asia aggravated by internal climate variability. *Nature Geoscience* 2023 16:2, 16(2), 154–161. <https://doi.org/10.1038/s41561-022-01111-0>

de Kok, R. J., Kraaijenbrink, P. D. A., Tuinenburg, O. A., Bonekamp, P. N. J., & Immerzeel, W. W. (2020). Towards understanding the pattern of glacier mass balances in High Mountain Asia using regional climatic modelling. *Cryosphere*, 14(9), 3215–3234. <https://doi.org/10.5194/tc-14-3215-2020>

Kronenberg, M., van Pelt, W., Machguth, H., Fiddes, J., Hoelzle, M., & Pertziger, F. (2022). Long-term firn and mass balance modelling for Abramov Glacier in the data-scarce Pamir Alay. *Cryosphere*, 16(12), 5001–5022. <https://doi.org/10.5194/tc-16-5001-2022>

McCarthy, M., Meier, F., Fatichi, S., Stocker, B. D., Shaw, T. E., Miles, E., Dussailant, I., & Pellicciotti, F. (2022). Glacier Contributions to River Discharge During the Current Chilean Megadrought. *Earth's Future*, 10(10), e2022EF002852. <https://doi.org/10.1029/2022EF002852>

Pritchard, H. D., Farinotti, D., & Colwell, S. (2021). Measuring changes in snowpack SWE continuously on a landscape scale using lake water pressure. *Journal of Hydrometeorology*, 22(4), 795–811. <https://doi.org/10.1175/JHM-D-20-0206.1>

Yan, D., Xu, H., Lan, J., Zhou, K., Ye, Y., Zhang, J., An, Z., & Yeager, K. M. (2019). Solar activity and the westerlies dominate decadal hydroclimatic changes over arid Central Asia. *Global and Planetary Change*, 173, 53–60. <https://doi.org/10.1016/J.GLOPLACHA.2018.12.006>

Yao, M., Tang, H., Huang, G., & Wu, R. (2024). Interdecadal shifts of ENSO influences on Spring Central Asian precipitation. *Npj Climate and Atmospheric Science* 2024 7:1, 7(1), 1–11. <https://doi.org/10.1038/s41612-024-00742-x>

REVISION STATEMENT - Response to reviewers

Snowfall decrease in recent years undermines glacier health and meltwater resources in the Northwestern Pamirs.

by

Achille Jouberton, Thomas E. Shaw, Evan S Miles, Marin Kneib, Stefan Fugger, Pascal Buri, Michael McCarthy, Abdulhamid Kayumov, Hofiz Navruzshoev, Ardamehr Halimov, Khusrav Kabutov, Farrukh Homidov, Francesca Pellicciotti.

GENERAL REVISION

We would like to thank the editor and reviewers for this last round of reviews of our manuscript entitled “Snowfall decrease in recent years undermines glacier health and meltwater resources in the Northwestern Pamirs”, considered for publication in *Communications Earth & Environment*. We have revised the manuscript, and provide below a brief summary of the modifications.

Reviewer #1 gave their agreement for this manuscript to be published. Reviewer #3 and #4 both acknowledge our revisions addressing the comments from the first reviewers and find the manuscript acceptable for publication. They have provided mostly minor suggestions which we have all implemented in the revised version of the manuscript.

In response to their comments, we have:

- Added in the result section numbers that provide the absolute values of snow and icemelt water changes; these show that the increase in icemelt in the most recent period cannot compensate for snowmelt decrease and thus provide further evidence to support the manuscript title.
- Modified the map in Figure 1a, removing potentially contentious borders and replacing them with glacier regions outlines.
- Conducted an additional statistical analysis to further demonstrate that snowfall decline was driven by precipitation decline, and added a figure in the Supporting Information.
- Corrected the inconsistency in the study period (1999-2023 instead of 2000-2023) and checked accordingly the numbers calculated for these periods.
- Homogenized the formatting of references.
- Implemented the stylistic and language suggestions where appropriate.
- Reviewed the manuscript formatting guidelines and editorial requests table, and implemented modifications where needed.

We would like to thank the reviewers for their constructive comments, which further contributed to improving the manuscript. We very much hope that the revised

manuscript is now appropriate for publication in *Communications Earth & Environment*.

REVIEWER #1:

Agree to publish

REVIEWER #3:

General comments

The Pamir region's glaciers have attracted considerable interest owing to their unique, near-neutral mass balances amidst ongoing global change. Yet, the complex topography and limited observational data in this area have historically hindered a comprehensive understanding of the driving mechanisms behind glacier mass balance. This research endeavors to address this by leveraging in-situ observations, climate reanalysis, and remote sensing data to drive a land-surface model. This methodology enables the reconstruction of glacier changes from 2000 to 2023 and the identification of their underlying causes within a representative glacierized catchment in Tajikistan. This study is characterized by its meticulous detail, rigorous methodology, and substantial scientific merit. Furthermore, the authors have provided comprehensive and refined responses to feedback from prior reviewers. I deem this to be a highly commendable study and offer only a few minor suggestions for the authors' consideration prior to acceptance.

We thank the reviewer for their positive and constructive comments, which we believe have improved the quality and clarity of the manuscript. We respond to the specific comments below. Line or figure numbers refer to the revised manuscript unless otherwise stated.

Specific comments

1-Line 67 and Line 202: In Line 67, the authors state that "snowfall and snow cover have been substantially lower since 2018, leading to enhanced ice melt and affecting glacier mass balance and health." Conversely, in Line 202, it's mentioned that "The glacier melt contribution to runoff increased from 19% to 30% between the two sub-periods." My interpretation is that the period of reduced snowfall post-2018 is indeed associated with increased glacier melt and a larger proportion of meltwater contributing to runoff. However, the paper's title, "Snowfall decrease in recent years undermines glacier health and meltwater resources in the Northwestern Pamirs," suggests that decreased snowfall undermines meltwater resources. This seems contradictory. Could this be related to the "glacier melt inflection point," where glacier runoff initially increases before declining? I would appreciate clarification on this point.

We would like to thank the reviewer for sharing their view about this apparent contradiction. The reviewer is correct that the glacier runoff is increasing initially, and that it will likely decline in the future as the glacier retreats under continued warming.

As we state in Line 68-70 of the original manuscript (“The unbalanced ice melt resulting from a lack of snow compensated for a third of the precipitation-driven catchment runoff deficit, but accelerated glacier demise”), this additional ice melt is unbalanced firstly because it is not compensated by accumulation and leads to glacier mass loss, and secondly because it is not sufficient to compensate for all of the runoff reduction due to precipitation decrease. When ice melt and snow melt are considered together (on and off-glacier), which corresponds to the meltwater term in the title of the manuscript, there is an overall decrease (by - 46 mm w.e.) between 1999-2018 and 2018-2023. This means that even if ice melt did increase, it is of smaller magnitude than the snowmelt decrease, such that meltwater decreased as a result of snowfall decline and as indicated in the title. We have added a sentence in the result section (L271-275) to give these numbers, which complement well the relative contribution numbers already given and should prevent any misunderstanding. It reads as follows: “In absolute terms, the decrease in annual snowmelt (- 132 mm) is larger than the increase in ice melt (+86 mm), leading to an overall decrease in meltwater (-46 mm)”

To summarize, meltwater resources are undermined in the short term due to snowfall decrease, but also in the long term, as this reduction in glacier volume will eventually decrease the glacier’s ability to buffer precipitation-driven runoff deficit.

2-Figure 1(a) contains an incorrect map of China, specifically showing missing territory in the southwestern part of Xinjiang. Please replace it with the correct, standard map of China, which can be found at <http://bzdt.ch.mnr.gov.cn/>.

We would like to thank the reviewer for raising this point. We understand that some borders in this region are contentious, and certainly do not intend to represent a border favoring one country or another.

To avoid this issue, we have decided to remove country boundaries from Figure 1 (a), and to show instead the glacier outlines and mountain regions delineation from the Randolph 6.0 inventory, which are commonly used in maps of other publications in the field of glacio-hydrology. In addition, this allows us to indicate the location of the Pamir mountains, which was not explicitly shown in the original version of the figure.

3- Please define the meaning of "H-year" in the caption for Figure 3(b-d).

We have implemented this suggestion as follows: “H-year in panels b) to d) refers to the hydrological year (October 1st to September 30th)” (Line 1427-1429).

4-The statement in Line 110, "snowpack depletion was mainly caused by thinner winter snowpacks, and secondly by warmer spring conditions," appears to be slightly misphrased. Based on your results, it seems more accurate to state: "This shows that the snowpack's earlier melt-out from 2018 onwards is mostly controlled by the lower winter snowfall and snow height, but this is enhanced by slightly warmer spring air temperatures." Please revise for clarity.

We thank the reviewer for pointing out that this sentence can still be made clearer. Following their suggestion, we have revised it as follows: “Correlations between the simulated

snowpack melt-out date and the reconstructed climatic forcing (Fig. S41) show that the earlier seasonal snowpack melt-out from 2018 onwards was caused primarily by lower winter snowfall, leading to thinner winter snowpacks, and secondarily by warmer spring air temperatures (+0.19°C, Fig. S36)” (Line 175-183).

5-The authors repeatedly emphasize that reduced snowfall is primarily driven by precipitation changes, a crucial conclusion highlighted in the abstract. While Figures S37 and S38 illustrate precipitation variations, I suggest adding two scatter plots: one showing the linear regression between temperature and snowfall, and another between precipitation and snowfall. Comparing the R-squared values of these two scatter plots could more effectively demonstrate the dominant role of precipitation in the observed snowfall changes in the Pamir region.

As pointed out correctly by the reviewer, we emphasized that the reduced snowfall was primarily driven by precipitation changes, because the snowfall to total precipitation ratio did not change noticeably, as shown in Figure 4b.

Following the reviewer’s suggestion, we have added the two scatter plots in the Supplementary Material (Fig. S43) in Section 8 (Linear regression analyses). The interpretation of the R-squared values is straightforward: 82% of the variance in annual snowfall is explained by annual precipitation, while only 2% is explained by annual air temperature. This further demonstrates the dominant role of precipitation in the snowfall changes reported in our manuscript.

We mention this new figure in the manuscript Lines 194-198 as follows: “Snowfall changes cannot be attributed to a change in the annual snowfall fraction, which remained stable between the two periods (Fig. 4b), but rather to precipitation changes (Fig. S43).”

Figure R1/S43: Linear regression between the annual snowfall and annual precipitation (a), and between annual snowfall and mean annual air temperature (b). These variables correspond to catchment-wide averages. The number given in the top-left corner of each panel is the coefficient of determination. Each dot represents a hydrological year.

6-In Line 351, "automatic weather stations" should be replaced with "AWSs," as this

abbreviation has already been defined earlier in the text.

We have implemented this suggestion, thanks.

7-References: Formatting Inconsistencies. There are numerous inconsistencies in the reference list. For instance, some references include DOIs while others do not, and journal names are sometimes abbreviated inconsistently. Additionally, some references provide excessive information (e.g., lines 547-551). Please ensure that all references adhere consistently to the journal's formatting guidelines or at least maintain internal consistency throughout the list.

We have now homogenized the reference formatting in the revised version of the manuscript, consistently with the journal's guidelines.

REVIEWER #4:

General comments:

The manuscript has already been reviewed by two other reviewers before me and therefore the manuscript is in very good shape. After reading the comments of reviewer#1 and response from the authors, I feel the authors have attempted to respond well with as much as possible explanations (from the literatures, for such as: reasons/drivers for reduced snowfall in recent period and futuristic perspective of the glaciers/snowfall in the region) and additional analysis. I also agree with the authors that some of the explanations reviewer#1 asked for (e.g., reasons/drivers for reduced snowfall and futuristic perspective of the glaciers/snowfall) are difficult to point out based on the study objectives/tasks the authors set and in the literature such large-scale change understanding has not yet been investigated/understood well. Therefore, in my opinions, the explanations and modifications by the authors are good and sufficient considering their focus of the study, which is exploring the glacier changes in the region using modelling and possible explanations of the drivers and conditions.

Below I point out some of the small changes, majorly stylistic and language related, that needs to be fixed before its acceptance.

We thank the reviewer for their positive and constructive comments, which we believe have improved the quality and clarity of the manuscript. We are also thankful that they acknowledge the explanations and modifications we implemented following the comments of former reviewer #1 and deem them sufficient. We respond to their specific comments below. Line or figure numbers refer to the revised manuscript unless otherwise stated.

Specific comments

Abstract

I think opening sentence, L 22-24 (in the marked_up manuscript file), is a bit too long and the last part of it is not well paired with the first part of it. What I try to mean is, the first part reads ‘..world’s last relatively healthy mountain glaciers..’ which sounds positive, however, the last part reads ‘..causes of this anomalous glacier state are not known.’ Which sounds negative but the different behaviour of glaciers here are not presented before, so it sounds like where does the ‘anomalous glacier state’ come from. I would recommend rephrasing the sentence and dividing into two parts maybe for better read.

We agree with the reviewer that the different behaviour of glaciers are not presented here, and cannot be due to the word limits. Therefore we have changed the word “anomalous” by “stable”.

L 27-29, snowfall should be placed before snow depth which is more logical.

We agree with the reviewer and have made the corresponding change: “We show that snowfall and snow depth have been substantially lower since 2018.” (Line 54)

L 29-31, does the authors want to rephrase it to ‘..cause of higher/increased mass losses..’ or just ‘mass losses’ is fine? – because I see that the recent years mass losses are higher than the decade let’s say 2000s (which the authors already mention in the next line, L 31-32).

We thank the reviewer for this suggestion. While it is true for Kyzylsu Glacier that the mass losses are higher since 2018 than in the 2000s, we would prefer to leave ‘mass losses’ as it is. This sentence is about regional implications, and using “higher” or “increased” might not be true for all glaciers in the region, as some did not experience mass loss in the early 2000s but began losing mass only in more recent years.

Introduction

L 48-51, here the authors says that their modelling period is 1999-2023 (also in L64 later), but in abstract it was mentioned as 2000-2023. Please check. Additionally, I would also quickly point out the study catchment name here, may be in this way ‘..in Tajikstan (Kyzylsu catchment),..’ because until reading, the readers will already be curious to know about the site. Moreover, the appearance of the site name in the next para is bit sudden. Therefore, I feel mentioning quickly in L48-51 would be a good choice.

We would like to thank the reviewer for pointing out this inconsistency between the two periods of 1999-2023 and 2000-2023. The simulation period is indeed from 1 October 1999 to 30 September 2023. We have changed 2000 to 1999 where necessary in the manuscript. We consistently checked all calculations for the whole period and sub-periods (1999-2018 vs 2018-2023) and modified the numbers where necessary. The modifications of some numbers are however very minor.

Methods

1. L 328-329, here I think ‘while’ is not necessary and reads stylistically awkward. Instead, ‘therefore’ may be used.

The classification of these glaciers as winter-accumulation type was not made from precipitation data but rather from glacier surface observations (cf. Huang et al. 2022 which is cited there), such that using “therefore” here would make an incorrect link. We replaced ‘while’ with ‘and’, which reads more neutral and hopefully less awkward.

Results

1. L 93-95, I think this sentence belongs to the Methods section, somewhere. It sounds a bit sudden to appear with less context for it. Please check if the authors can place it somewhere in the Methods section. The hydrological year has already been mentioned in the ‘land-surface model’ sub-section in Methods below

We agree that the definition of the hydrological year should be in the Methods, and have removed the definition of the hydrological year from here, as it was already mentioned in the Methods sub-section, as correctly pointed out by the reviewer.

We agree that the specification of the standard deviation might feel out of place, but we believe that it is more helpful for the reader to have this at the beginning of the result section, close to where it is relevant, rather than in any of the method sub-sections. In this way, we prevent the risk that a reader might see these ranges as an indication of uncertainty.

Review of the manuscript "**Snowfall decrease in recent years undermines glacier health and meltwater resources in the Northwestern Pamirs**" written by Jouberton et al.

The manuscript has already been reviewed by two other reviewers before me and therefore the manuscript is in very good shape. After reading the comments of reviewer#1 and response from the authors, I feel the authors have attempted to respond well with as much as possible explanations (from the literatures, for such as: reasons/drivers for reduced snowfall in recent period and futuristic perspective of the glaciers/snowfall in the region) and additional analysis. I also agree with the authors that some of the explanations reviewer#1 asked for (e.g., reasons/drivers for reduced snowfall and futuristic perspective of the glaciers/snowfall) are difficult to point out based on the study objectives/tasks the authors set and in the literature such large-scale change understanding has not yet been investigated/understood well. Therefore, in my opinions, the explanations and modifications by the authors are good and sufficient considering their focus of the study, which is exploring the glacier changes in the region using modelling and possible explanations of the drivers and conditions.

Below I point out some of the small changes, majorly stylistic and language related, that needs to be fixed before its acceptance.

Abstract

1. I think opening sentence, L 22-24 (in the marked_up manuscript file), is a bit too long and the last part of it is not well paired with the first part of it. What I try to mean is, the first part reads ‘..world’s last relatively healthy mountain glaciers..’ which sounds positive, however, the last part reads ‘..causes of this anomalous glacier state are not known.’ Which sounds negative but the different behaviour of glaciers here are not presented before, so it sounds like where does the ‘anomalous glacier state’ come from. I would recommend rephrasing the sentence and dividing into two parts maybe for better read.
2. In L 27-29, snowfall should be placed before snow depth which is more logical.
3. L 29-31, does the authors want to rephrase it to ‘..cause of higher/increased mass losses..’ or just ‘mass losses’ is fine? – because I see that the recent years mass losses are higher than the decade let’s say 2000s (which the authors already mention in the next line, L 31-32).

Introduction

1. L 48-51, here the authors says that their modelling period is 1999-2023 (also in L64 later), but in abstract it was mentioned as 2000-2023. Please check. Additionally, I would also quickly point out the study catchment name here, may be in this way ‘..in Tajikstan (Kyzylsu catchment),..’ because until reading, the readers will already be curious to know about the site. Moreover, the appearance of the site name in the next para is bit sudden. Therefore, I feel mentioning quickly in L48-51 would be a good choice.

(before going through the results/discussion, I was more curious to learn about the methods, therefore, I present my opinions on Methods before Results)

Methods

1. L 328-329, here I think 'while' is not necessary and reads stylistically awkward. Instead, 'therefore' may be used.

Results

1. L 93-95, I think this sentence belongs to the Methods section, somewhere. It sounds a bit sudden to appear with less context for it. Please check if the authors can place it somewhere in the Methods section. The hydrological year has already been mentioned in the 'land-surface model' sub-section in Methods below.

The Pamir region's glaciers have attracted considerable interest owing to their unique, near-neutral mass balances amidst ongoing global change. Yet, the complex topography and limited observational data in this area have historically hindered a comprehensive understanding of the driving mechanisms behind glacier mass balance. This research endeavors to address this by leveraging in-situ observations, climate reanalysis, and remote sensing data to drive a land-surface model. This methodology enables the reconstruction of glacier changes from 2000 to 2023 and the identification of their underlying causes within a representative glacierized catchment in Tajikistan.

This study is characterized by its meticulous detail, rigorous methodology, and substantial scientific merit. Furthermore, the authors have provided comprehensive and refined responses to feedback from prior reviewers. I deem this to be a highly commendable study and offer only a few minor suggestions for the authors' consideration prior to acceptance.

1-Line 67 and Line 202: In Line 67, the authors state that "snowfall and snow cover have been substantially lower since 2018, leading to enhanced ice melt and affecting glacier mass balance and health." Conversely, in Line 202, it's mentioned that "The glacier melt contribution to runoff increased from 19% to 30% between the two sub-periods." My interpretation is that the period of reduced snowfall post-2018 is indeed associated with increased glacier melt and a larger proportion of meltwater contributing to runoff. However, the paper's title, "Snowfall decrease in recent years undermines glacier health and meltwater resources in the Northwestern Pamirs," suggests that decreased snowfall **undermines** meltwater resources. This seems contradictory. Could this be related to the "glacier melt inflection point," where glacier runoff initially increases before declining? I would appreciate clarification on this point.

2-Figure 1(a) contains an incorrect map of China, specifically showing missing territory in the southwestern part of Xinjiang. Please replace it with the correct, standard map of China, which can be found at <http://bzdt.ch.mnr.gov.cn/>.

3- Please define the meaning of "H-year" in the caption for Figure 3(b-d).

4-The statement in Line 110, "snowpack depletion was mainly caused by thinner winter snowpacks, and secondly by warmer spring conditions," appears to be slightly misphrased. Based on your results, it seems more accurate to state: "This shows that the snowpack's earlier melt-out from 2018 onwards is mostly controlled by the lower winter snowfall and snow height, but this is enhanced by slightly warmer spring air temperatures." Please revise for clarity.

5-The authors repeatedly emphasize that reduced snowfall is primarily driven by precipitation changes, a crucial conclusion highlighted in the abstract. While Figures S37 and S38 illustrate precipitation variations, I suggest adding two scatter plots: one showing the linear regression between temperature and snowfall, and another between precipitation and snowfall. Comparing the R-squared values of these two scatter plots could more effectively demonstrate the dominant role of precipitation in the observed snowfall changes in the Pamir region.

6-In Line 351, "automatic weather stations" should be replaced with "AWSs," as this abbreviation has already been defined earlier in the text.

7-References: Formatting Inconsistencies. There are numerous inconsistencies in the reference list. For instance, some references include DOIs while others do not, and journal names are sometimes abbreviated inconsistently. Additionally, some references provide excessive information (e.g., lines 547-551). Please ensure that all references adhere consistently to the journal's formatting guidelines or at least maintain internal consistency throughout the list.